# Western US intraplate deformation controlled by the complex lithospheric structure

Zebin Cao [1,2] & Lijun Liu [1,2] ✉

The western United States is one of Earth's most tectonically active regions, characterized by extensive crustal deformation through intraplate earthquakes and geodetic motion. Such intracontinental deformation is usually ascribed to plate boundary forces, lithospheric body forces, and/or viscous drag from mantle flow. However, their relative importance in driving crustal deformation remains controversial due to inconsistent assumptions on crustal and mantle structures in prior estimations. Here, we utilize a fully dynamic three-dimensional modeling framework with data assimilation to simultaneously compute lithospheric and convective mantle dynamics within the western United States. This approach allows for quantitative estimations of crustal deformation while accounting for the realistic three-dimensional lithospheric structure. Our results show the critical role of the complex lithospheric structure in governing intraplate deformation. Particularly, the interaction between the asthenospheric flow and lithospheric thickness step along the eastern boundary of the Basin and Range represents a key driving mechanism for localized crustal deformation and seismicity.

The driving mechanism behind intraplate deformation remains a fundamental scientific question. One example is the western United States (WUS), which displays a broad and complex crustal deformation pattern (Fig. 1). Since the mid-Cenozoic, much of the WUS has been experiencing long-lasting crustal extension, with well-documented geological and geophysical evidence[1]. However, the forces driving the extensional tectonics remain debated, with proposed mechanisms broadly falling into three categories: long-range plate boundary forces[2–4], lithospheric body forces resulting from lateral gradients of gravitational potential energy (GPE)[2–8], and basal tractions exerted by the underlying mantle flow[5,7,9–13]. Many researchers realized that a single mechanism is insufficient to explain the observed crustal deformation[1–5,7,9–11,14], but the relative importance of different mechanisms is still controversial. One key outstanding question is how the three-dimensional (3D) lithospheric structure, a feature usually simplified in previous studies, is linked to the proposed driving forces.

Contemporary intraplate deformation in the WUS is evident through the observed crustal motion deviating from rigid plate rotation and widespread intraplate earthquakes. In the WUS, Global Positioning System (GPS) measurements revealed a clockwise rotation pattern[15] in lateral crustal motion relative to the stable North American plate[16] (Fig. 1a). Generally, the Basin and Range (B&R) is continuously extending in the northwest direction[17,18], while the Great Valley (GV) and Colorado Plateau (CP) remain relatively undeformed[17–20]. Notably, the crustal motion within the southern B&R is lower than that further north[21]. From the northwestern B&R to the Pacific Northwest (PNW), the crustal motion transitions from northwest to northeast[15,22]. This change in the surface motion was primarily attributed to the changing driving force, transitioning from right-lateral shear along the Pacific-North American plate boundary to northeastern push from subduction of the Juan de Fuca (JdF) plate[15]. While this interpretation aligns with observed surface kinematics, the broader deformation over the entire WUS may also reflect other driving forces, such as those from lateral

[1]State Key Laboratory of Lithospheric Evolution, Institute of Geology and Geophysics, Chinese Academy of Sciences, Beijing, China. [2]Department of Earth Science & Environmental Change, University of Illinois at Urbana-Champaign, Urbana, IL, USA. ✉e-mail: ljliu@mail.iggcas.ac.cn

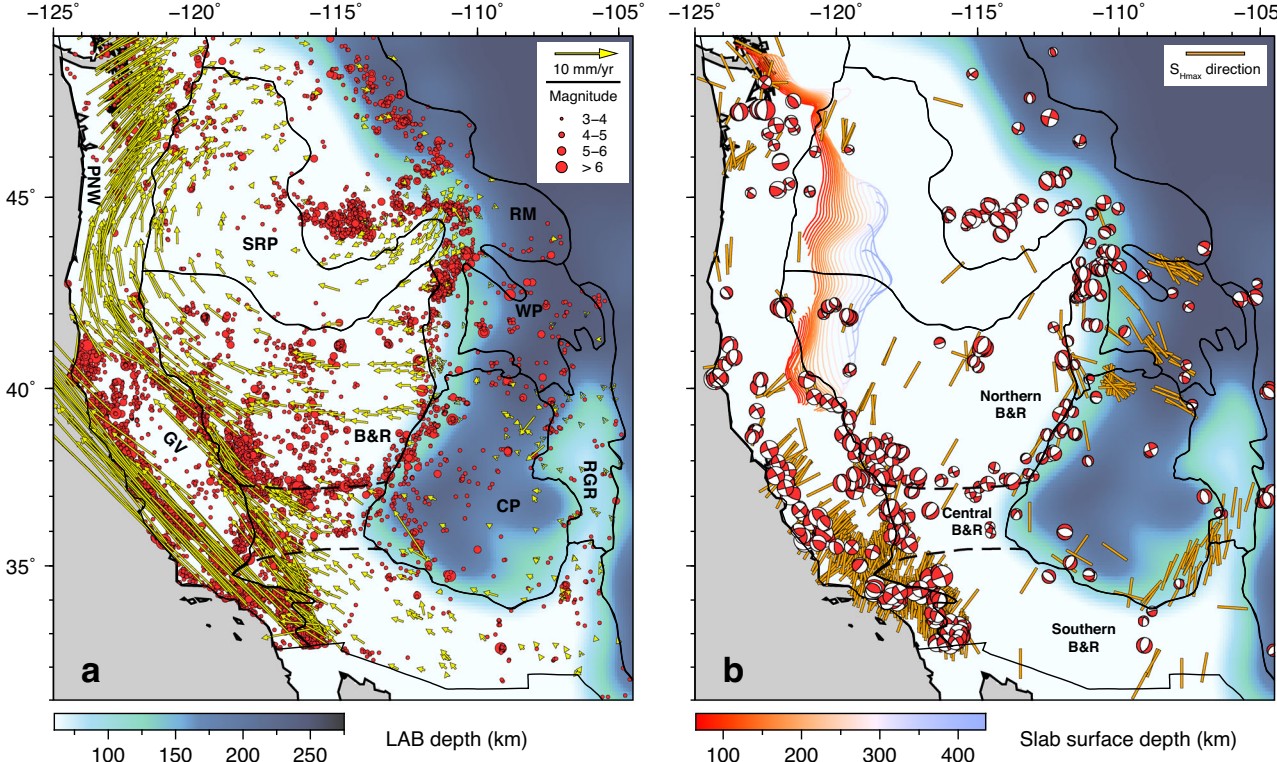

**Fig. 1 | Observed crustal deformation features and stress state in the WUS.**
**a** Seismicity and geodetically measured crustal motion overlying the LAB depth ("Methods" section). Red dots indicate earthquake locations from 1980 to 2020, with the radius showing magnitudes (from USGS). Yellow arrows indicate crustal motion relative to the stable North American plate (in the NAM14 reference frame[16]) determined with GPS measurements (from GAGE/UNAVCO[89]). **b** Slab surface[36,37] (the colored contours indicate depths), observed focal mechanism solutions[23,24], and geophysically estimated maximum horizontal compression ($S_{Hmax}$) directions[26] overlying the LAB depth. Black lines in both panels indicate major tectonic provinces in the WUS. The dashed lines within the Basin and Range divide it into the northern, central, and southern segments, following ref. 90. PNW Pacific Northwest, GV Great Valley, SRP Snake River Plain, B&R Basin and Range, RM Rocky Mountain, WP Wyoming Plateau, CP Colorado Plateau, RGR Rio Grande Rift.

gradients of lithospheric GPE and active mantle flow, whose exact roles remain unclear.

The seismicity in the WUS correlates with the geodetically measured crustal motion (Fig. 1a) and can be categorized into four main belts/regions: the San Andreas Fault (SAF), Walker Lane (WL), Intermountain Seismic Belt (ISB), and PNW. Recent geophysical studies revealed that the widespread seismicity in the WUS is strongly associated with the spatially varying crustal stress state[23–26] (Fig. 1b). The relatively undeformed GV is bounded by the SAF and WL with prominent seismicity, characterized mainly by strike-slip faulting (Fig. 1b). The remarkable ISB, a north-south-trending seismic belt stretching from the southern CP to the northern Rocky Mountain (RM)[27,28], coincides with the eastern boundary of B&R as well as the transition from thin to thick North American lithosphere (Fig. 1a). Most earthquakes in the ISB show normal faulting with E-W extension along the eastern boundary of B&R and the tectonic parabola surrounding the Yellowstone hot spot track, but the crustal stress transitions to a more compressional state in the northern RM with earthquakes exhibiting strike-slip to thrust faulting (Fig. 1b). Seismicity in the PNW is spatially distributed, mainly in the back-arc region, which corresponds to the diffuse crustal motion in this area.

The debated mechanisms for WUS crustal deformation could be boiled down to the often simplified and yet uncertain lithospheric and convective mantle structures adopted in different studies and the associated fine-scale dynamics[2–14,29–37]. Particularly, the lithosphere–asthenosphere interaction below this region remains contentious. Previous studies generally assumed a flat lithosphere–asthenosphere boundary (LAB) for the continental plate

at 100 km or deeper[2–7,9–11,13]. Under this assumption, the convective mantle and lithospheric dynamics were often computed separately and connected through one-way coupling that only allowed the convecting mantle to exert horizontal shear (i.e., viscous drag) along the flat LAB[4,6,7,11,12]. This horizontal shear was usually estimated from mantle convection models that only considered long-wavelength deep density anomalies[4,6,7,11,12]. Consequently, this approach may not adequately resolve the fine-scale interaction between the convective mantle and the lithosphere due to the lack of spatially varying lithospheric thickness and associated mantle deformation[35–37]. Indeed, the assumed flat LAB contradicts recent global[38,39] and regional[40–42] studies showing significant lateral variations in lithospheric thickness.

In addition, the importance of 3D lithospheric effective viscosity structure, which impacts stress transmission and crustal deformation, has been previously undervalued. Earlier studies either assumed mechanically strong continental plates with uniform strength[9,11–13] or focused solely on lateral variations in bulk lithospheric effective viscosity[2–5,7,10–12]. In the latter scenario, lithospheric dynamics was frequently modeled under the thin-shell approximation, treating the lithosphere as a thin two-dimensional (2D) layer with constant thickness and vertically uniform properties. This method usually estimated lateral viscosity variations from geodetically or geologically determined deformation fields[2,3,5,7,10]. However, recent studies highlighted the crucial need for independently determined 3D lithospheric effective viscosity structures to accurately model crustal deformation[31]. Hence, a more sophisticated modeling approach beyond the thin-shell approximation is essential to fully capture the detailed 3D lithospheric dynamics.

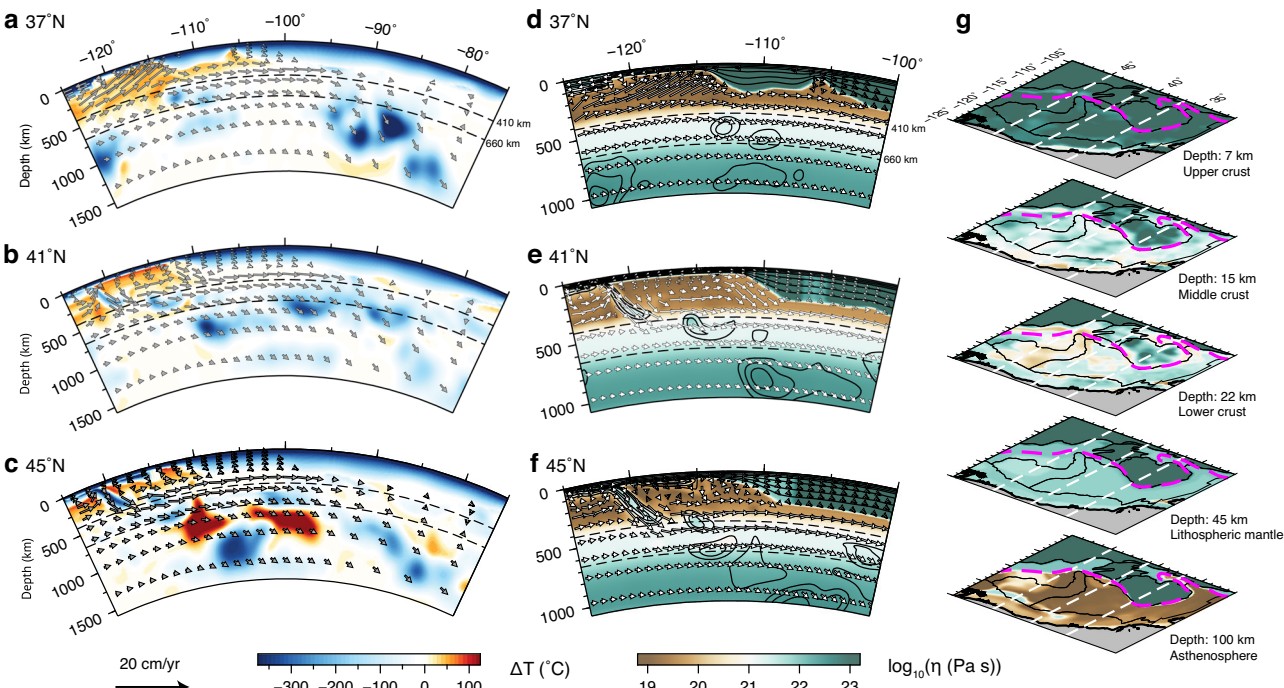

**Fig. 2 | Lithospheric and convecting mantle structures in the fully dynamic 3D model. a–c** Cross-sections showing the thermal structure (adapted from refs. [36,37]) beneath the North American plate at different latitudes. **d–f** Cross-sections showing the effective viscosity structure within the WUS at different latitudes. **g** Map views of the 3D effective viscosity structure within the WUS. The arrows in (**a**)–(**f**) show the mantle flow relative to the stable North American plate. The black contours in (**d**)–(**f**) are isotherms at every 100 °C. In (**g**), the magenta dashed lines mark the location of LAB at 75 km, approximating the transition from thin to thick lithosphere in LTM1, and the white dashed lines show the locations of cross-sections (**a**)–(**f**).

Here, we present a set of data-orientated 3D dynamic models with detailed lithospheric and convective mantle processes computed within a unified physical framework. The best-fitting fully dynamic model well reproduces the observed surface topography (Supplementary Fig. 1), directional pattern of crustal stress field[25,26] (Supplementary Fig. 2), surface deformation rates[43] (Supplementary Fig. 3), and GPS-measured differential crustal motion. Utilizing the best-fitting model, we provide quantitative analyses on crustal deformation in the tectonically active WUS and the role of the realistic 3D lithosphere.

## Results and discussion
### Construction of the fully dynamic 3D model
We model the lithosphere and convective mantle as an incompressible viscous fluid in spherical geometry ("Methods" section). The numerical model covers a region spanning $60° \times 100° \times 2890$ km in latitude × longitude × radius, with the finest resolution (~7 km) inside the lithosphere and asthenosphere within the WUS. The model's high resolution allows us to faithfully simulate the fine-scale 3D dynamics in the WUS, especially the previously unexplored lithosphere–asthenosphere interaction arising from the spatially varying lithospheric thickness and shallow mantle flow. Unlike most previous studies that employed one-way coupling, our model incorporates natural coupling between the convecting mantle and lithosphere by simultaneously including the realistic lithospheric and convecting mantle structures (Fig. 2). Furthermore, the data-oriented nature of our geodynamic modeling approach allows us to test models with different lithospheric and convecting mantle structures to isolate their respective effects on crustal deformation. In this study, we construct ten different models (Table 1) to fully explore the driving mechanism for WUS intracontinental deformation.

In our model ("Methods" section), the present-day mantle structure is adopted from a time-dependent mantle flow model using the hybrid geodynamic modeling approach[44], which combines the forward[45] and adjoint[46] data-assimilation algorithms. The present-day 3D JdF slab structure was reproduced by the high-resolution forward approach[45], and other mantle structures, including the subducted Farallon slab, lithospheric drips, and hot-mantle anomalies, were derived from recent high-resolution seismic tomography[42,47] via the adjoint method[46]. This time dependence matches several independent geophysical and geological observations within the WUS. For example, the evolving mantle thermal state reproduced the late-Cenozoic volcanic history[36] and topographic changes[48]. The associated mantle flow and density anomalies also matched the SKS splitting measurements[37] and free-air gravity anomaly[48].

The density and viscosity structures of the continental lithosphere, characterized by strong lateral variations in thickness, play a critical role in truthfully modeling lithospheric deformation[2–4,6,7,14,31]. We employ a hybrid approach that merges LITHO 1.0[38] and a recent regional body-wave tomography model[42] to construct our lithospheric thickness model ("Methods" section, LTM1), which effectively captures the first-order lithospheric thickness change along the eastern boundary of B&R (Fig. 1). Using the seismically detected Moho depth[49] and LAB depth[38,42], we deduce the lithospheric density structure by matching surface topography ("Methods" section). Additionally, we construct a 3D lithospheric effective viscosity structure based on recent seismic attenuation studies[50,51] ("Methods" section, Fig. 2g). This approach allows us to independently calculate the fine-scale lithospheric deformation by avoiding direct reliance on GPS measurements.

### Lithosphere–asthenosphere interaction beneath the WUS
Our fully dynamic 3D model well captures the fine-scale interaction between the lithosphere and convecting mantle beneath the WUS. The dominantly eastward upper-mantle flow in the region is mainly driven by the sinking Farallon slab currently located beneath the central and eastern US[36,37] (Fig. 2a–f). Locally, the flow is diverted by the

**Table 1 | Model setup and comparison with observation**

| Model Number | Moho depth[a] | LAB depth[b] | Crustal properties | | Lithospheric mantle properties | | Convecting mantle structures | Fit to observation | | | | Magnitude of velocity residual |
|---|---|---|---|---|---|---|---|---|---|---|---|---|
| | | | Density[c] | Viscosity | Density | Viscosity | | $S_{Hmax}$[d] | $S_{Hmax}$[e] | Seismicity | Angular misfit to surface motion | |
| 1 | 30 km | LTM2 | 2850 kg/m³ | 5 × 10²² Pa s | 3340 kg/m³ | 5 × 10²² Pa s | All included | 23.1° | 23.2° | 34.33% | 38.1° | 4.01 mm/yr |
| 2 | Variable | LTM2 | The preferred crustal density model | 5 × 10²² Pa s | 3340 kg/m³ | 5 × 10²² Pa s | All included | 23.3° | 26.2° | 35.50% | 51.7° | 4.44 mm/yr |
| 3 | Variable | LTM1 | The preferred crustal density model | The preferred 3D crustal viscosity structure | The preferred lithospheric mantle density model | The preferred 3D lithospheric mantle viscosity structure | All included | 25.4° | 21.9° | 96.37% | 37.8° | 1.78 mm/yr |
| 4 | Variable | LTM1 | N/A | The preferred 3D crustal viscosity structure | 3340 kg/m³ | The preferred 3D lithospheric mantle viscosity structure | None | 23.9° | 22.4° | 89.88% | 44.4° | 1.93 mm/yr |
| 5 | Variable | LTM1 | N/A | The preferred 3D crustal viscosity structure | 3340 kg/m³ | The preferred 3D lithospheric mantle viscosity structure | All included | 30.6° | 25.4° | 96.62% | 45.1° | 2.40 mm/yr |
| 6 | Variable | LTM1 | The preferred crustal density model | The preferred 3D crustal viscosity structure | The preferred lithospheric mantle density model | The preferred 3D lithospheric mantle viscosity structure | None | 22.8° | 23.8° | 95.22% | 40.6° | 2.07 mm/yr |
| 7 | N/A | LTM2 | N/A | 5 × 10²² Pa s | 3340 kg/m³ | 5 × 10²² Pa s | None | 22.8° | 27.9° | 73.10% | 46.7° | 4.50 mm/yr |
| 8 | Variable | LTM2 | N/A | The preferred 3D crustal viscosity structure | 3340 kg/m³ | 5 × 10²² Pa s | None | 22.2° | 23.2° | 88.89% | 43.4° | 2.53 mm/yr |
| 9 | N/A | LTM2 | N/A | 5 × 10²² Pa s | 3340 kg/m³ | 5 × 10²² Pa s | All included | 23.6° | 25.2° | 78.11% | 34.8° | 4.13 mm/yr |
| 10 | Variable | LTM2 | N/A | The preferred 3D crustal viscosity structure | 3340 kg/m³ | 5 × 10²² Pa s | All included | 27.2° | 23.6° | 96.94% | 34.0° | 2.53 mm/yr |

[a]The term "variable" represents the Moho depth model shown in Supplementary Fig. 6a, while "N/A" indicates that the crust is not included in the model.
[b]LTM1 features a geophysically inferred, laterally varying LAB depth ("Methods" section), whereas LTM2 maintains a flat LAB at a depth of 100 km.
[c]The preferred crustal density structure has a constant density of 2850 kg/m³ across the entire model domain, except for the SRP, which has a density of 2950 kg/m³ due to its enrichment in basaltic composition. "N/A" indicates that the crustal density anomaly is not considered in the model.
[d]Comparison to Levandowski et al.[25].
[e]Comparison to Lund Snee and Zoback[26].

subducting JdF slab and the thick cratonic lithosphere on the east (Fig. 3a). Below the region with a thin lithosphere, the predicted mantle flow pattern independently matches the recent observation of asthenospheric azimuthal anisotropy[52] (Fig. 3a). The modeled highly segmented JdF slab, consistent with seismic tomography[53], allows hot asthenospheric material to flow through the central slab tear and around its edges to beneath the northern B&R and southern SRP[36,37], respectively. Further inland, the eastward flow is bisected by the thick CP lithosphere, where the southern branch of hot mantle flows around the CP into the Rio Grande Rift (RGR), and other hot material flows northeastward until blocked by the thick cratonic lithosphere east of the B&R. This peculiar flow pattern is vividly tracking the azimuthal anisotropy below the thin WUS lithosphere, with an average angular misfit of 37.5° between the directions of flow and anisotropy ("Methods" section). In the cratonic region, our model predicts little internal deformation within the lithospheric mantle. This implies that there is no active shearing within the lithosphere to generate prominent azimuthal anisotropy. Therefore, we interpret the sudden change of anisotropy orientation across the eastern boundary of B&R as reflecting a contrast between active asthenospheric deformation on the west and frozen-in fabric in the cratonic lithospheric mantle to the east.

To illustrate the importance of lateral variations in lithospheric thickness on WUS mantle dynamics, we analyze the asthenospheric flow using different lithospheric thickness models (Fig. 3b). The predicted asthenospheric flow exhibits near-identical patterns with a laterally varying (LTM1; red arrows in Fig. 3b) and a constant (LTM2; light-blue arrows in Fig. 3b) lithospheric thickness beneath the region with a thin lithosphere. However, the fast horizontal flow with LTM2 continues to go eastward beneath the intermountain west,

while the flow in the model with LTM1 largely stops here (Fig. 3b). Notably, the fast eastward mantle flow predicted with LTM2 is nearly orthogonal to the observed azimuthal anisotropy beneath the intermountain west, resulting in a large average angular misfit of 49.7°. Although seismic anisotropy does not uniquely reveal the pattern of mantle flow, it could help eliminate certain scenarios. For example, the flow predicted with LTM2 (Fig. 3b) suggests strong E-W azimuthal anisotropy east of the B&R, opposite to observation (Fig. 3a). In contrast, the flow with LTM1 implies little deformation in this region, not violating the anisotropy constraint that could reflect fossil fabric instead.

## Complex lithospheric structure controls WUS crustal deformation

The E-W contrast in WUS lithospheric thickness (Fig. 1) implies a complex 3D effective viscosity and density structure. In the thin lithosphere region on the west, both the crust and lithospheric mantle are rheologically weak (Fig. 2d–g) and actively deforming (Fig. 1a). In contrast, the cratonic lithosphere east of B&R is much thicker and less tectonically active; this thick root blocks and diverts shallow asthenospheric flow (Fig. 3a). Besides, recent studies demonstrated that the thick cratonic lithospheric mantle is denser than the ambient asthenosphere and modulates the crustal stress and surface topography[14,54–56]. Despite these findings, the role of 3D lithospheric structure in modulating crustal deformation remains largely unexplored.

To quantitatively assess the impact of 3D lithospheric structure on crustal deformation, we conduct three simulations (M1-M3) using different lithospheric structures with increasing complexity. The crustal deformation rate (i.e., the second invariant of strain rate tensor)

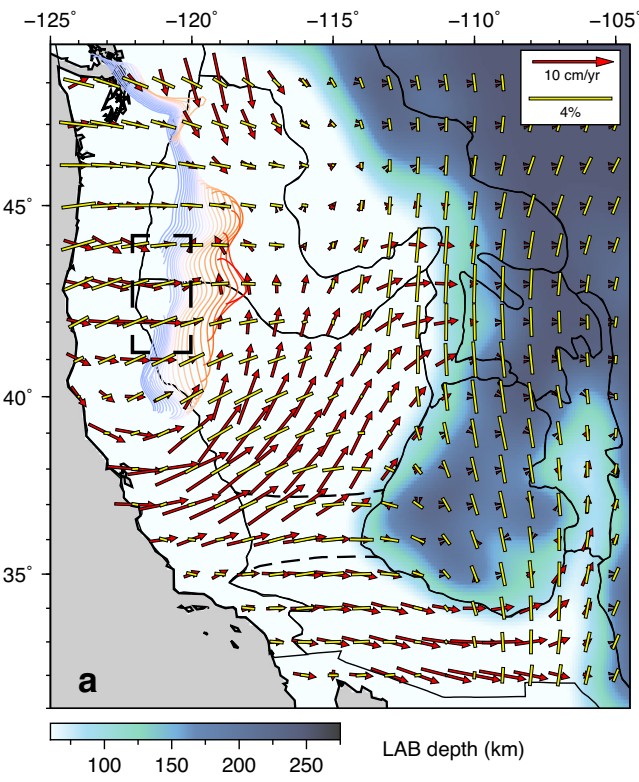

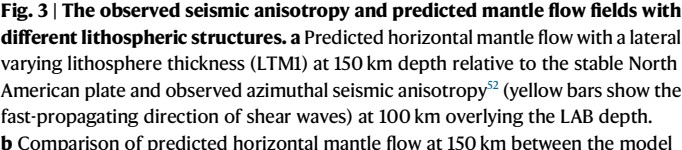

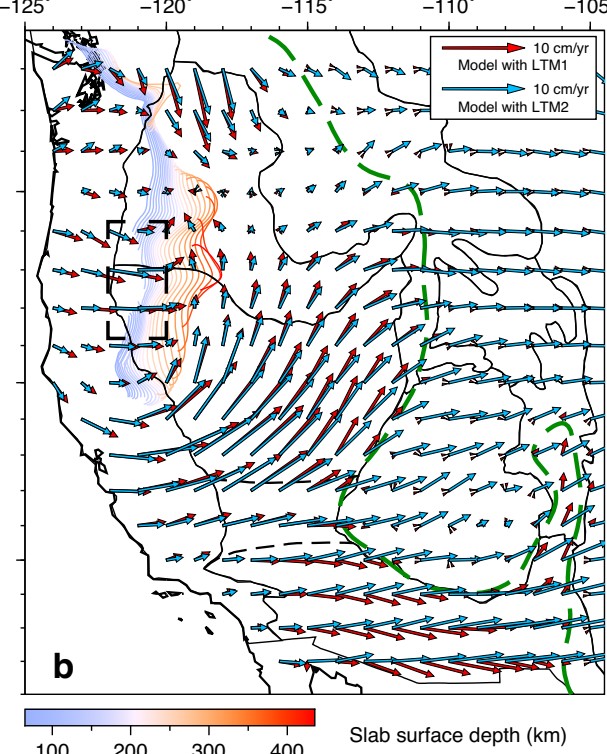

**Fig. 3 | The observed seismic anisotropy and predicted mantle flow fields with different lithospheric structures. a** Predicted horizontal mantle flow with a lateral varying lithosphere thickness (LTM1) at 150 km depth relative to the stable North American plate and observed azimuthal seismic anisotropy[52] (yellow bars show the fast-propagating direction of shear waves) at 100 km overlying the LAB depth. **b** Comparison of predicted horizontal mantle flow at 150 km between the model

with a lateral varying lithosphere thickness (LTM1) and the model with a constant lithosphere thickness (LTM2). In both panels, the colored contours indicate the upper surface of the JdF slab[36,37] at different depths, and the dashed black boxes represent a slab tear, which allows hot asthenospheric material to flow through. The green dashed line in (**b**) marks the location of LAB at 75 km, approximating the lithospheric thickness step in LTM1.

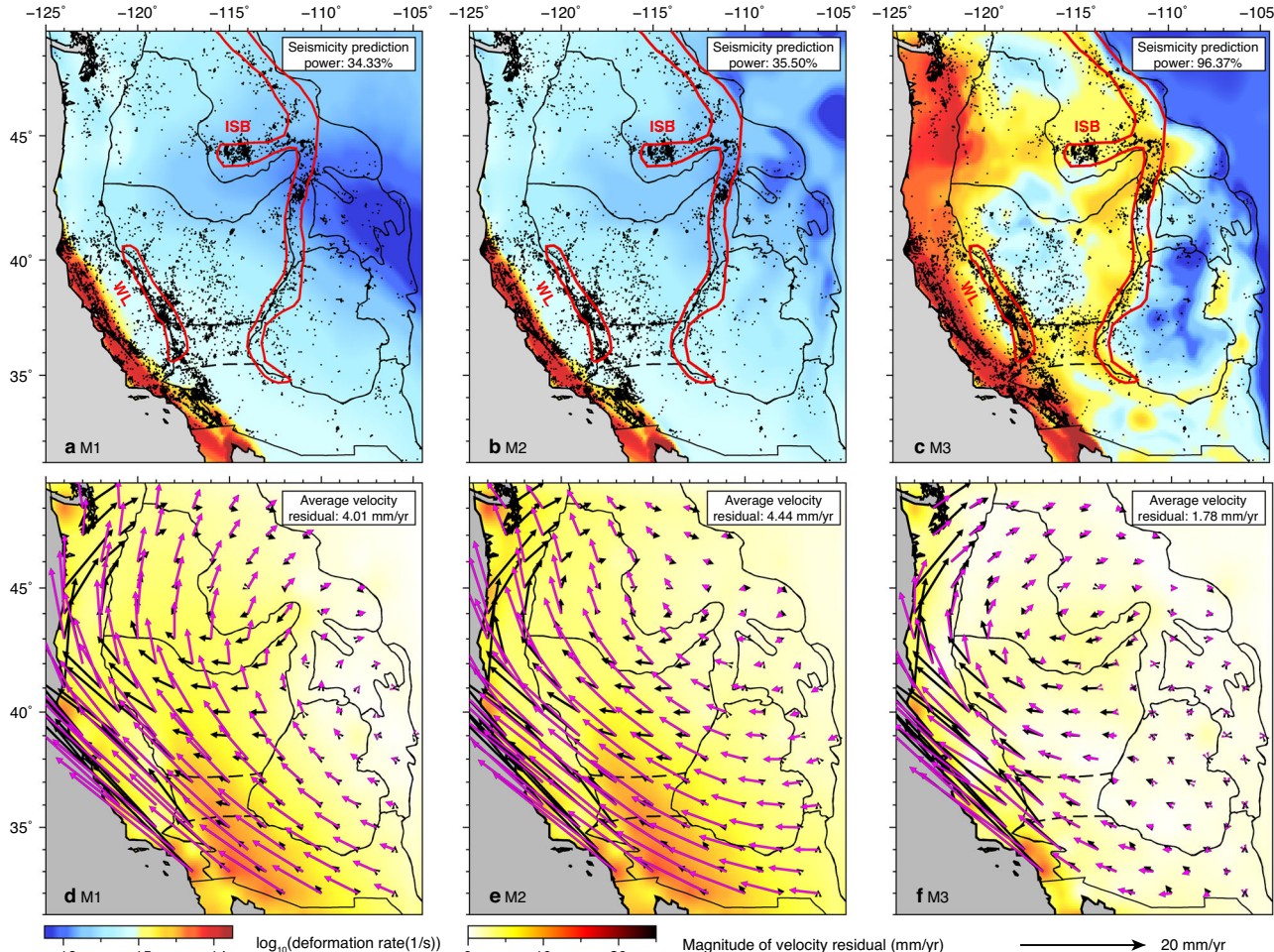

**Fig. 4 | Predicted crustal deformation with different lithospheric structures.** Predicted depth-averaged crustal deformation rate (the second invariant of strain rate tensor) in the uppermost 30 km with **a** a uniform lithosphere, M1, **b** a lithosphere of constant thickness but varying Moho depth, M2, **c** a heterogeneous lithosphere as geophysically inferred, M3. Comparisons between the GPS-measured surface motion and predicted surface motion from **d** M1, **e** M2, and **f** M3. Black dots in (**a**)–(**c**) indicate earthquake locations from 1980 to 2020. The thick red lines outline the WL and ISB in (**a**)–(**c**). In (**d**)–(**f**), the black bars represent the smoothed GPS measurements, and the magenta bars represent the predictions.

and seismicity are expected to be correlated[57–59]. Therefore, we compare the predicted crustal deformation rate to observed seismicity, while employing a seismicity prediction power analysis ("Methods" section) to quantify the match between model prediction and observation, with a higher value representing a better match. To compare with the GPS measurements, we calculate the vectorial mismatch between the predicted surface motion and a smoothed version of GPS-determined surface motion ("Methods" section).

The model with the simplest lithospheric structure (M1), consisting of a laterally uniform crust and lithospheric mantle, predicts a diffusive deformation pattern without clear localization of strain rate (Fig. 4a). The seismicity prediction power of M1 is 34.33%. Most modeled deformation is concentrated along the plate boundary (i.e., the SAF). The predicted crustal motion, whose magnitude decays inland (Fig. 4d), is dominated by the plate boundary effect (Supplementary Fig. 4a–c). The predicted surface motion deviates from observation by an average residual of 4.01 mm/yr.

In the second model (M2, Fig. 4b, e), we further consider the lateral gradients of crustal GPE, another primary driving force for intraplate deformation[3,8,60]. We approximate this GPE using laterally varying Moho depth[60] and a constant crustal density at 2850 kg/km³, except for the SRP, whose density is assumed to be 2950 kg/m³ due to its enrichment in basaltic composition[61,62]. Other model properties in

M2 are the same as in M1. The resulting crustal deformation rate is slightly elevated in regions with thick crust, such as the CP and Wyoming Plateau (WP), due to locally elevated GPE (Fig. 4b). Additionally, the high GPE in the intermountain west drives westward crustal motion in the WUS (cf. Fig. 4d, e). M2 has a seismicity prediction power of 35.50% and an average residual in crustal motion of 4.44 mm/yr, both similar to M1. Clearly, neither M1 nor M2 reproduces the observed seismicity distribution with localized strain accumulation and crustal motion. These tests call for a heterogeneous lithosphere, likely with laterally varying thickness and 3D variations in density and effective viscosity structures[14,31,54–56], to fully reproduce the observed crustal deformation field.

By incorporating the geophysically inferred lithospheric structure (Figs. 1 and 2), the resulting crustal deformation pattern correlates much better with the observed seismicity distribution (M3, Fig. 4c) and crustal motion (Fig. 4f), with the seismicity prediction power dramatically increasing to 96.37% and the average crustal motion residual dropping to 1.78 mm/yr. In particular, the WL and ISB display very high deformation rates, consistent with their intense seismicity. In contrast, the GV and CP have low deformation rates, also agreeing with seismicity and geodetic measurements[17–20]. The high deformation rate in the PNW extends far inland to the western SRP, supported by the distributed seismicity in this region. In addition, the observed

rotational crustal motion pattern in the WUS is well reproduced in this model (Fig. 4f). Therefore, the heterogeneous lithosphere is important for forming the observed crustal deformation pattern.

## Estimations of different driving forces and associated deformation

The estimations of proposed driving forces, namely plate boundary forces, lithospheric body forces, and basal tractions, depend on the adopted lithospheric structures. In our model, the plate boundary effect is defined as forces exerted on the continental plate by the moving oceanic plates with no contribution from the underlying mantle flow. Traditionally, basal tractions were defined as the horizontal shear exerted by mantle flow along a flat LAB[4,6,7,11,12]. However, in reality, the LAB exhibits undulations (Fig. 1) and dynamically interacts with the convecting mantle (Fig. 3a), resulting in complex surface tractions comprising both shear and normal forces[4]. Thus, in our model, we define basal tractions as the total surface forces along the undulating LAB, originating from the underlying mantle flow.

The conventional estimation of lithospheric body forces relied on computing lateral gradients of lithospheric GPE, assuming lithospheric isostasy with surface topography fully supported by lithospheric density anomalies above a presumed LAB depth of 100 km[2,3,7]. However, this assumed global compensation depth might not be appropriate, as the convective mantle also contributes to surface topography via radial stress along the LAB (i.e., dynamic topography). Furthermore, recent seismic studies revealed strong lateral gradients in LAB depth[38–42], particularly in regions like the CP and WP (Fig. 1), where the lithosphere exceeds 100 km in thickness. Consequently, the traditional calculation of lithospheric GPE is physically inaccurate. In our fully dynamic 3D model, lithospheric body forces refer to those originating from lithospheric density anomalies down to the undulating LAB. Unlike previous studies that take surface topography as an input for lithospheric GPE calculation[2,3,5,7–12], we use it as an independent observation to validate our lithospheric density structure ("Methods" section).

We construct seven additional geodynamic models (M4–M10, illustrated later) with different lithospheric and convective mantle structures (Table 1) to fully explore the impact of the heterogeneous lithosphere on crustal deformation and to quantitatively analyze the different driving forces. In all models, plate motion[63] is prescribed in the ocean basins, stable central-eastern US and Canada, as velocity boundary conditions ("Methods" section). Hence, the predicted deformation rate and surface motion include the plate boundary effect in all models.

Mantle convection influences lithospheric deformation through two mechanisms: the plate boundary effect, reflecting the integrated effect of global mantle flow, and basal tractions, arising from mantle flow right beneath the plate. To evaluate how a heterogeneous lithosphere impacts the contribution of mantle convection to crustal deformation, we perform six simulations and analyze the resulting crustal deformation patterns (Fig. 5 and Supplementary Fig. 5). In the models solely considering the plate boundary effect (M4, M7, and M8), we exclude all density anomalies in the convective mantle and continental lithosphere (Fig. 5a, d, Supplementary Fig. 5a, b, e, f). To further consider basal tractions, we utilize three models (M5, M9, and M10), each incorporating different lithospheric structures and all density anomalies in the convective mantle (Fig. 5b, e, Supplementary Fig. 5c, d, g, h). The crustal stress shown in Fig. 5b reflects the difference between M5 and M4, which have the same lithospheric structure but different convective mantle structures, thereby highlighting the contribution of basal tractions to crustal stress.

We first estimate the crustal stress and deformation resulting from the plate boundary effect by constructing M4, which assimilates a geophysically inferred heterogeneous lithosphere consisting of LTM1 and 3D variations in effective viscosity. Compared to the models

considering the plate boundary effect but having a constant lithospheric thickness (M7 and M8, Supplementary Fig. 5a, b), M4 has a larger seismicity prediction power (89.88%). Notably, along the lithospheric thickness step (the orange dashed line in Fig. 5a), the predicted crustal deformation is localized on the thin and weak side, with little crustal deformation in the interior of CP and WP (cf. Fig. 5a, Supplementary Fig. 5a, b). This finding highlights the importance of the lithospheric thickness step in generating localized crustal deformation. East of the lithospheric thickness step, the thick and strong lithospheric keel penetrates deeper into the asthenosphere, thereby enhancing the coupling between the thick lithosphere and the underlying static mantle. This locally enhanced coupling slows the northwestward motion of the thick lithospheric root (cf. Fig. 5d, Supplementary Fig. 5e, f, magenta arrows), effectively counteracting the shearing from the Pacific plate. As a result, the average residual reduces to 1.93 mm/yr.

In model M5, which further incorporates basal tractions, the predicted crustal deformation exhibits clearly localized high deformation rates in some intraplate regions, such as in the WL and ISB (Fig. 5b), with a seismicity prediction power of 96.62%. The inclusion of shallow hot anomalies (between the LAB and 100 km depth) that are not considered in cases incorporating basal tractions but having a flat LAB at 100 km (M9 and M10) leads to a reduction in asthenospheric effective viscosity (Fig. 2g). Consequently, the coupling between the lithosphere and convective mantle is locally weakened, resulting in a reduction in deformation rate in the interior of northern and southern B&R (cf. Fig. 5b, Supplementary Fig. 5c, d). In contrast, along the lithospheric thickness step (the orange dashed line in Fig. 5b), the crustal deformation rate is largely unchanged (cf. Fig. 5b and Supplementary Fig. 5d), indicating the interaction between the mantle flow and thick lithospheric keel generates localized crustal deformation along the step that counteracts the decoupling effect. Compared to the case without the density-driven mantle flow (M4, Fig. 5a), it shows that the interaction between the active mantle flow and thick lithospheric root locally increases the crustal deformation rate, like in the northern RM and along the eastern boundary of northern B&R (cf. Fig. 5a, b). The eastward mantle flow also pushes the lithospheric root eastwards, reducing westward motion in the southern CP and central and southern B&R (cf. Fig. 5e, Supplementary Fig. 5g, h).

To estimate crustal stress and deformation due to the lithospheric body forces over the WUS, we construct a model (M6) that incorporates a heterogeneous lithosphere with 3D viscosity and density variations. In this model, all density anomalies in the convective mantle are removed. The crustal stress shown in Fig. 5c is the difference between M6 and M4, representing the crustal stress solely due to the lithospheric body forces. The lithospheric body forces, arising from lateral gradients of lithospheric GPE, elevate the crustal deformation rate over the whole WUS (cf. Fig. 5a, c). While our estimated crustal stress due to the lithospheric body forces is generally consistent with previous studies[2–4,7–9], it is noteworthy that the underlying physics differs. We emphasize that a continental lithosphere with a laterally varying thickness and a denser-than-asthenosphere lithospheric mantle is critical in calculating and understanding the lithospheric GPE[14,56]. Additionally, the high GPE east of the B&R drives the WUS to move to the northwest, particularly in the B&R, SRP, and PNW (cf. Fig. 5d, f).

## Mechanism for localized intraplate deformation

While the 3D crustal effective viscosity structure influences the crustal deformation rate ("Methods" section), the primary driving force for the localized crustal deformation along the lithospheric thickness step is the locally enhanced interaction between the lithosphere and asthenosphere (Figs. 5 and 6). This enhanced lithosphere–asthenosphere interaction ultimately affects the crustal effective viscosity structure, particularly the intraplate crustal weak zone (Fig. 2g), which cannot be fully explained by other mechanisms

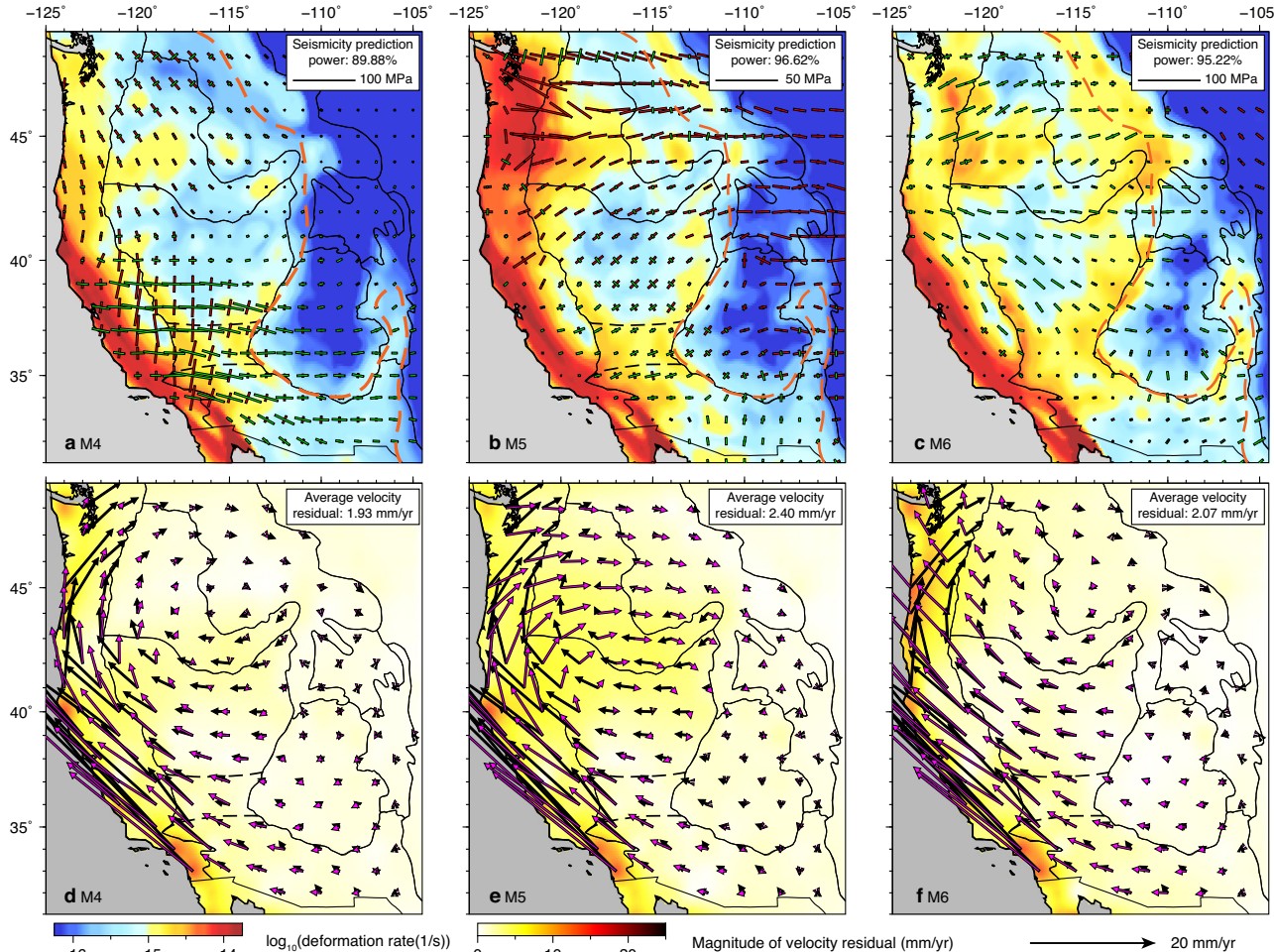

**Fig. 5 | Predicted crustal deformation and stress due to different driving forces.** Predicted crustal deformation rate and crustal stress in the uppermost 30 km from **a** M4 with LTM1, a 3D lithospheric effective viscosity structure, and a none convective mantle, **b** M5 with a lithosphere the same as M4 and all density anomalies in the convective mantle, **c** M6 with LTM1, a 3D lithospheric effective viscosity structure, all density anomalies in the lithosphere, and a none convective mantle. Comparisons between the GPS-measured surface motion and predicted surface motion from **d** M4, **e** M5, and **f** M6. In (**a**)–(**c**), the bars show the directions and magnitudes of horizontal principal stresses, with red representing compression and green representing extension. In (**a**)–(**c**), the orange dashed line marks the location of LAB at 75 km, approximating the lithospheric thickness step in LTM1. In (**d**)–(**f**), the black bars represent the smoothed GPS measurements, and the magenta bars represent the predictions.

("Methods" section). Our model demonstrates that beneath regions with a thin lithosphere, the mantle flow does not generate significant shear on the horizontal plane (Fig. 6a–c) due to the low asthenospheric effective viscosity (Fig. 2g), which is consistent with previous studies[6,7]. However, our model reveals a strong eastward normal traction (i.e., pressure drag) on N-S trending vertical planes beneath the region with a thin lithosphere (Fig. 6e, f), which is not previously captured. This increased eastward normal traction is due to the thick lithospheric keel blocking the landward mantle flow (M3, Figs. 3 and 6), as it leads to large positive dynamic pressure in the convecting mantle beneath the thin lithosphere region. This pressure drag (~20 MPa), which is part of basal tractions, pushes the lithospheric keel eastwards and increases the horizontal shear in regions with a thick lithosphere, such as the CP, WP, and northern RM (cf. Fig. 6a–c). Furthermore, this pressure drag drives localized lithospheric deformation along the lithospheric thickness step at the eastern boundary of B&R (Fig. 5a, b), representing a key mechanism for the ISB (Figs. 1a and 4).

Recent seismological studies[41] revealed fine-scale lithospheric thinning beneath the eastern boundary of northern B&R, a factor that is not included in our lithospheric thickness model. This observed lithospheric thinning may lead to further localization of lithospheric

deformation by reducing the bulk lithospheric strength, similar to the effect of a localized crustal weak zone. The shallow LAB observed beneath the eastern boundary of northern B&R appears anti-correlated with the crustal thickness, suggesting its formation as a result of recent tectonic activities[41]. Therefore, we propose that the localized interaction between hot asthenospheric flow and lithospheric thickness step, starting around 12 Ma[36], plays an important role in generating locally thinned lithosphere.

## Revised role of different forces in driving WUS crustal deformation

The importance of variable lithospheric properties (Figs. 3–6) requires a reconsideration of previously proposed driving forces (plate boundary forces, lithospheric body forces, and basal tractions) for WUS crustal deformation. In many previous studies, the estimations of driving forces were influenced by the simplified lithospheric structures adopted[2,3,5–7,10–12]. We have shown that the lithospheric thickness step creates a strong pressure drag (Fig. 6) that causes localized intraplate deformation (Figs. 4 and 5), a factor that has been previously neglected. Additionally, the presence of a dense and thick cratonic root alters the estimation of lithospheric GPE, which calls for a reevaluation of its contribution to crustal deformation. Our best-fitting model (M3) not

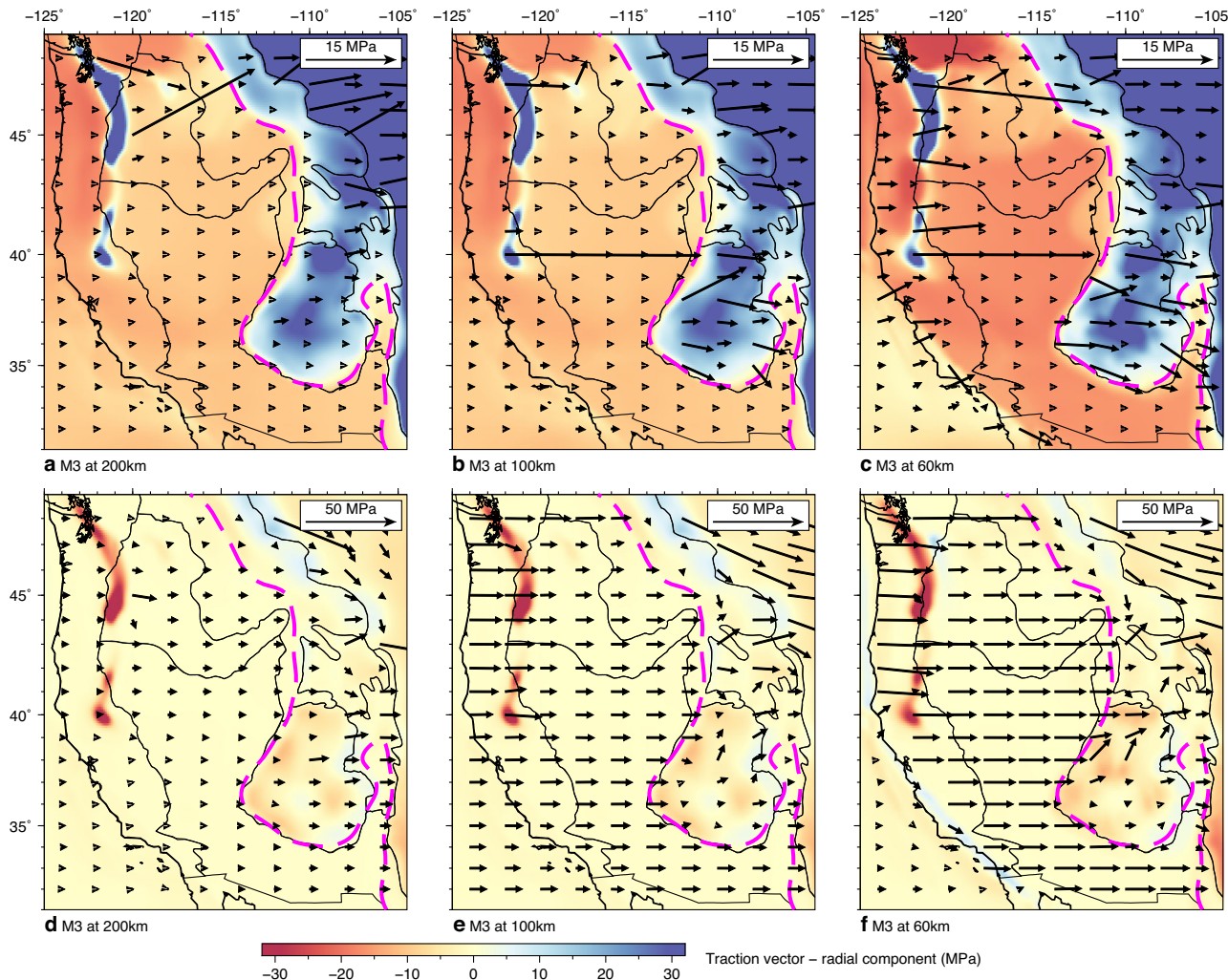

**Fig. 6 | Predicted traction field at different depths from the best-fitting model, M3. a–c** Tractions on horizontal planes. **d–f** Tractions on N-S trending vertical planes. In all panels, the background color shows the radial component of the traction vector, with red representing upward traction and blue representing downward traction. The vectors show the horizontal component. The magenta dashed line marks the location of LAB at 75 km, approximating the lithospheric thickness step in LTM1.

only successfully reproduces the observed crustal deformation (Fig. 4c, f), but also predicts a crustal stress field consistent with high-quality earthquake focal mechanism solutions[23,24] and recent geophysical estimations of the horizontal maximum compression directions ($S_{Hmax}$) and Aφ parameter[25,26] (Fig. 7). Based on the best-fitting model, we further evaluate the relative importance of plate boundary forces (Fig. 5a), basal tractions (Fig. 5b), and lithospheric body forces (Fig. 5c) by calculating the ratios of their magnitudes (i.e., the second invariant of stress tensor; Fig. 8).

The observed focal mechanism solutions[23,24], geophysically estimated $S_{Hmax}$ directions[25,26], and Aφ parameter[25,26] show strong spatial variations over the WUS that cannot be fully explained by one single driving force[9,14,25,26]. In the southwest coastal region, the plate boundary effect dominates the crustal motion and stress field (Figs. 4f, 5a, d, 7a and 8e, f). Along the SAF, the Pacific plate exerts right-lateral shearing forces, driving strike-slip faulting (Figs. 5a and 7a, c) and northwestward crustal motion (Fig. 5d). The PNW and northern SRP are complex regions with observed crustal stress generally in N-S compression[23–26] (Fig. 1b), which was largely attributed to the plate boundary forces from the shearing along the SAF and subduction of the JdF plate[15,25,64]. In our best-fitting model, all three driving mechanisms contribute large stresses in the PNW (Fig. 5a–c). The crustal stress generated by GPE gradients and basal tractions is generally in the same direction with

opposite signs in this region. Combined with the plate boundary forces exerted by the JdF plate and Pacific plate, the PNW and northern SRP experiences strong N-S to NW-SE compression (~20–40 MPa), with crustal stress state in the thrust faulting regime (Fig. 7c). Overall, the basal tractions play the most important role in deforming the PNW and northern SRP (Fig. 8d–f). Further inland, in the central and southern B&R, the westward extension (Fig. 7a, c) is predominantly driven by the plate boundary force originating from the Pacific-North American boundary with strong E-W extension and weak N-S compression (Fig. 5a). Furthermore, the N-S extensional stress due to basal tractions (Fig. 5b) and E-W extensional stress from the gradients of lithospheric GPE (Fig. 5c) collectively modify the crustal stress field into purely E-W extension, aligning with the focal mechanism solutions and geophysical estimations of $S_{Hmax}$ directions and Aφ parameter (Fig. 7). In the northern B&R, extensional crustal stresses due to plate boundary forces and GPE gradients generally have comparable magnitudes (Fig. 8f). In the northwestern B&R, the crustal stress field shows a combined effect of all three driving forces with comparable magnitudes (Fig. 8d–f). Further north, the E-W compression in the northern RM is predominantly from basal tractions (Fig. 8d, e), which are locally enhanced by the strong and neutrally buoyant slab curtain[65]. Noteworthy, along the eastern boundary of B&R and the tectonic parabola surrounding the Yellowstone hot spot track, where the ISB is located, the GPE-induced

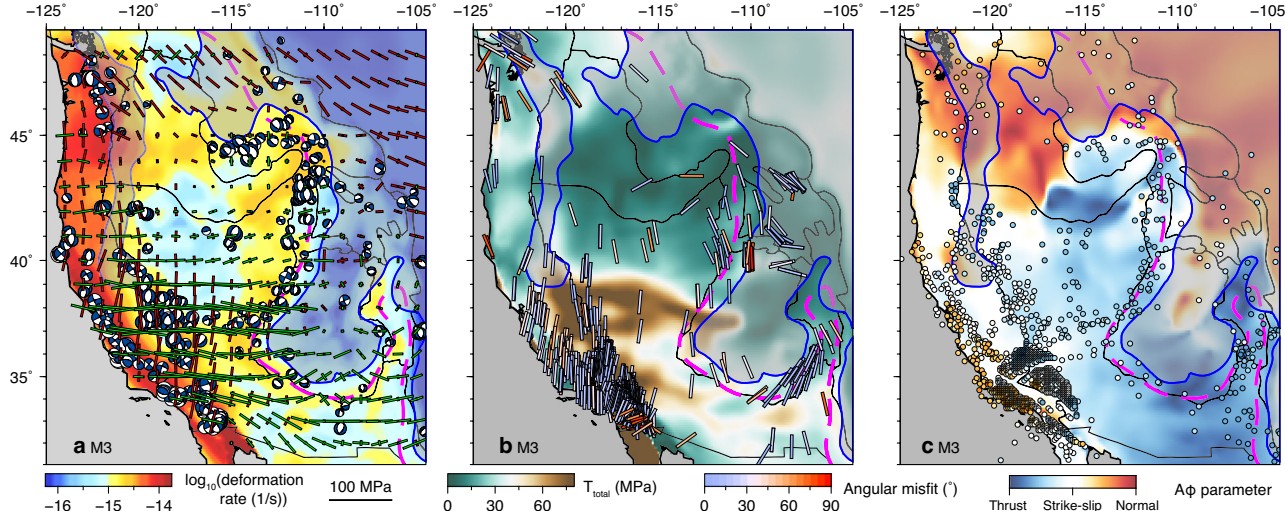

**Fig. 7 | Comparisons between the predicted crustal stress field from M3 and geophysical observations. a** Predicted crustal stress and observed focal mechanism solutions[23,24] overlying the predicted depth-averaged crustal deformation rate. **b** Predicted maximum horizontal compression directions ($S_{\text{Hmax}}$) overlying the predicted second invariant of crustal stress tensor ($T_{\text{total}}$), with the bars showing the predicted $S_{\text{Hmax}}$ directions and color showing angular misfit to the geophysical estimations[25]. **c** Geophysical estimations of Aφ parameter[25] (dots) overlying the predicted Aφ parameter. The shaded regions with blue outlines represent cold materials, including subducting slab, slab curtain, lithospheric drips, and continental lithosphere, at 100 km (with the outline representing a temperature 10 °C cooler than the ambient mantle).

extensional stress dominates the crustal stress field and drives normal to strike-slip faulting (Figs. 7a and c, 8d, f), in line with the focal mechanism solutions[23,24]. Overall, in the WUS, the plate boundary effect controls the coastal region and the central-southern B&R, while lithospheric body forces dominate the northern B&R, and basal tractions resulting from mantle flow play an important role in deforming the northern WUS (Figs. 8 and 9).

Our systematic analysis demonstrates the crucial role of a 3D heterogeneous lithosphere fully coupled with the underlying convecting mantle in accurately reproducing observed crustal deformation. In particular, the lateral variation in lithospheric thickness significantly influences the contribution of mantle convection to crustal deformation, as well as the estimation and understanding of lithospheric body forces (Figs. 5, 9, and Supplementary Fig. 5). The GPS-measured lateral surface deformation in the WUS reflects a combined effect of plate boundary forces, lithospheric body forces, and basal tractions. Among all driving forces, only basal tractions drive eastward differential motion over the WUS, a crucial factor in reproducing the observed surface deformation (Figs. 4f and 5d–f). This finding highlights the importance of shallow mantle flow interacting with a lithosphere that has laterally varying thickness (Figs. 3 and 4), as is often neglected in previous studies. Furthermore, the heterogeneous lithospheric structure plays a critical role in generating the observed crustal deformation within the WUS. Particularly, the lithospheric thickness step along the eastern boundary of B&R is essential in generating localized intraplate deformation, which leads to the formation of the enigmatic ISB (Figs. 1 and 6).

## Methods

### Data-oriented geodynamic modeling approach

To construct a self-consistent, fully dynamic model with coupled lithosphere and convective mantle, we generally follow the modeling approach described in Cao and Liu[14]. The lithosphere and mantle are modeled as an incompressible viscous fluid in spherical geometry under the Boussinesq approximation. We use a user-modified version of CitcomS with tracers[14,66,67] to solve the momentum equation.

The high-resolution 3D models cover a region of 60° × 100° × 2890 km in latitude × longitude × radius, with the finest resolution (~7 km) inside the lithosphere and asthenosphere within the WUS. This model domain is sufficiently larger than the conterminous US. It contains the surrounding ocean basins and parts of Canada and Mexico, allowing natural modeling of the detailed 3D lithospheric and mantle dynamics, including the fine-scale lithosphere–asthenosphere interaction that was not captured in previous studies.

A hybrid velocity boundary condition is used on the top of the model to incorporate far-field forces and truthfully simulate intra-continental deformation. Plate motion[63] is prescribed in the ocean basins surrounding the conterminous US, the stable central and eastern US (east of 110°E), and part of Canada (north of 55°N), while the actively deforming WUS and Mexico are set to have free-slip boundary condition. All other boundaries have a free-slip velocity boundary condition. Each model is an individual simulation with different lithospheric and mantle structures, as listed in Table 1.

### Lithospheric and convective mantle structures

The convective mantle structure is adopted from a time-dependent mantle flow model using the hybrid geodynamic modeling approach[44]. This time-dependent convecting mantle structure matches multiple geophysical and geological observations[36,37,48]. Most features in the convecting mantle, including the subducted Farallon slab, lithospheric drips, and hot asthenospheric material, were derived from high-resolution seismic tomography models[42,47] using the adjoint assimilation scheme[46]. Features near the subduction zone, including the subducting Juan de Fuca slab and mantle wedge, were generated by a forward sequential assimilation scheme[45]. The convecting mantle has a temperature- and depth-dependent effective viscosity structure, the same as Zhou et al.[36,37].

The lithosphere structure in our model is inferred from recent seismic studies[38,42,49–51,68–70]. The initial lithospheric thickness model has a Moho depth model from CRUST 1.0[49] and a hybrid LAB depth model[14] derived from LITHO 1.0[38] and a body-wave tomography model[42]. Since LITHO 1.0[38] does not capture the abrupt lithospheric thickness change along the eastern boundary of the B&R, which is better imaged in the recent regional body-wave seismic tomography[42], we redefine this part of the LAB depth based on the

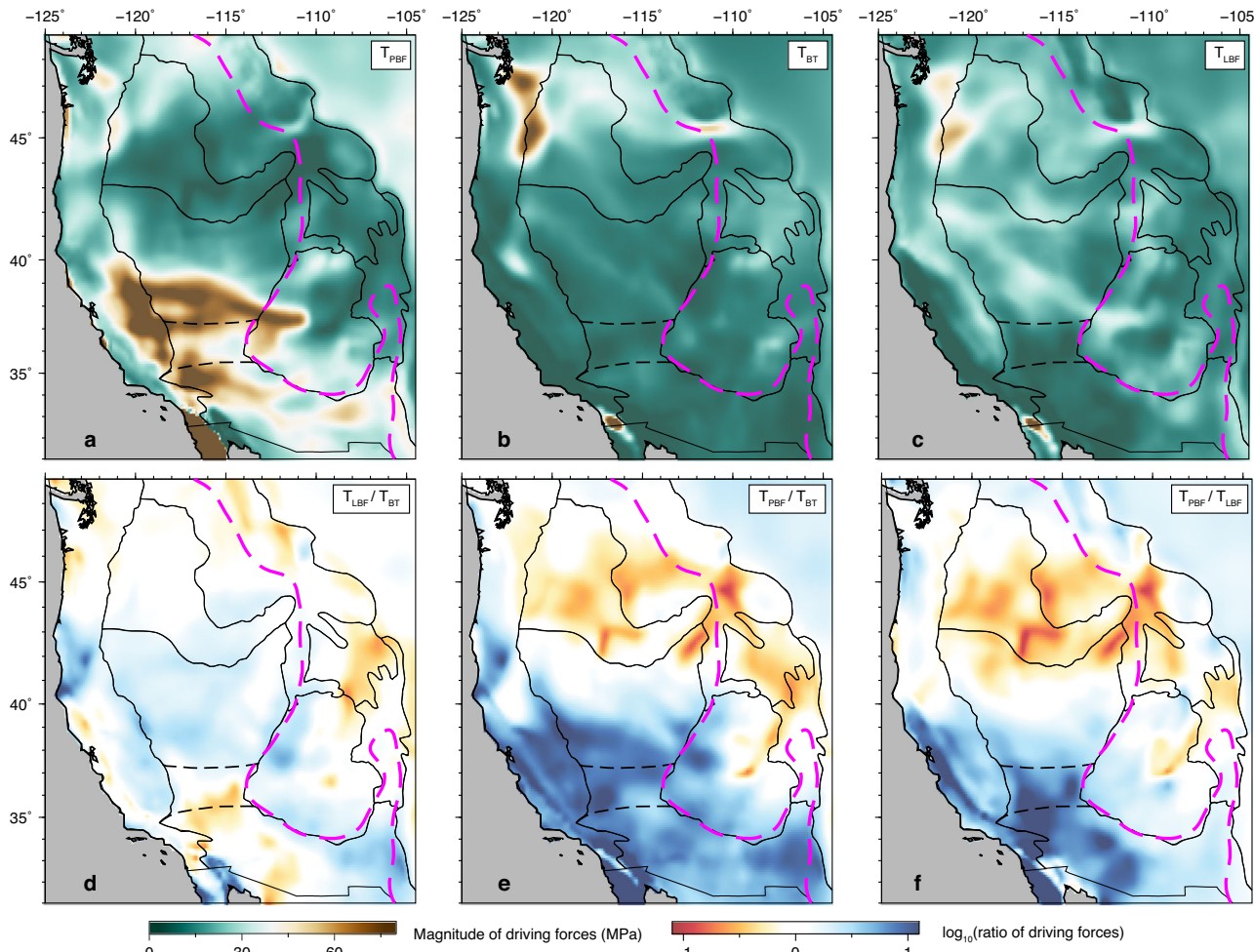

**Fig. 8 | The magnitude and relative importance of proposed driving forces in the WUS.** The second invariant of the crustal stress tensor caused by **a** plate boundary forces ($T_{PBF}$), **b** basal tractions ($T_{BT}$), and **c** lithospheric body forces ($T_{LBF}$). Ratios of driving forces with **d** $T_{LBF}/T_{BT}$, **e** $T_{PBF}/T_{BT}$, and **f** $T_{PBF}/T_{LBF}$. In (**d**)–(**f**), red indicates the greater importance of the force in the denominator, white denotes that the two forces are comparable, and blue shows the force in the numerator is more important.

body-wave seismic tomography model. In practice, we assume the continuous fast anomalies down to 250 km from the seismic tomography model represent the continental lithosphere. We also adopt an MLD in the thick cratonic lithospheric mantle at 120 km, based on recent seismic studies[68–70]. Seismic studies revealed that the MLD marks the boundary between two layers with different seismic properties[68–70], also indicating changes in their physical properties. Besides, recent geophysical and geodynamic studies showed that the continental lithospheric mantle has a layered density structure[14,54–56,71–73]. Therefore, we assume the MLD also marks an interface for lithospheric mantle densities.

The initial lithospheric density structure has three layers. In the crust, we assume a constant density of 2850 kg/m³, except for the SRP. In the SRP, its lower-than-ambient elevation was attributed to its basalt-rich crust[61,62], so we adopt a crustal density of the SRP at 2950 kg/m³, which is the same as that in Zhou and Liu[48]. The density of the lithospheric mantle is elusive, and the proposed density ranges from neutrally buoyant[74–76] to denser than ambient asthenospheric material[14,54–56,71–73,77]. In our lithosphere model, the density structure of the lithospheric mantle has two layers separated by the seismically observed MLD[68–70]. The densities of these two layers were adjusted to reproduce the continental-scale surface topography contrast between the WUS and the central and eastern US, as well as between the Pacific Ocean and the WUS[14]. The resultant best-fitting

lithospheric mantle density structure has an upper layer with a density of 3355 kg/m³ over the whole conterminous US and a lower layer with a density of 3375 kg/m³ in the central US and 3385 kg/m³ in the eastern US, respectively. This lithospheric mantle density structure is consistent with recent geophysical estimations[56,72,73], which showed the cratonic lithospheric mantle is overall ~1% denser than ambient asthenosphere (3340 kg/m³).

To capture the fine-scale lithospheric GPE variations, we fix the density for each layer and region, then adjust the LAB and Moho depth to reproduce the surface topography within the WUS. This process involves two steps: (1) calculate the initial topography difference between the observation and prediction, then adjust the LAB depth to accommodate the initial topography difference with constraints detailed in Cao and Liu[14]; (2) calculate the residual topography difference between the observation and new prediction from step 1, then adjust the Moho depth to accommodate the residual topography difference. In each step, we first conduct a simulation with the unmodified Moho and LAB depth to predict the surface topography. In the simulation, all density anomalies in the convecting mantle are included, so dynamic topography is naturally included in the prediction. This modeling approach does not rely on the assumption of isostasy. The resulting Moho depth is generally consistent with the recent seismically determined Moho depth in the WUS[49,78] (Supplementary Fig. 6).

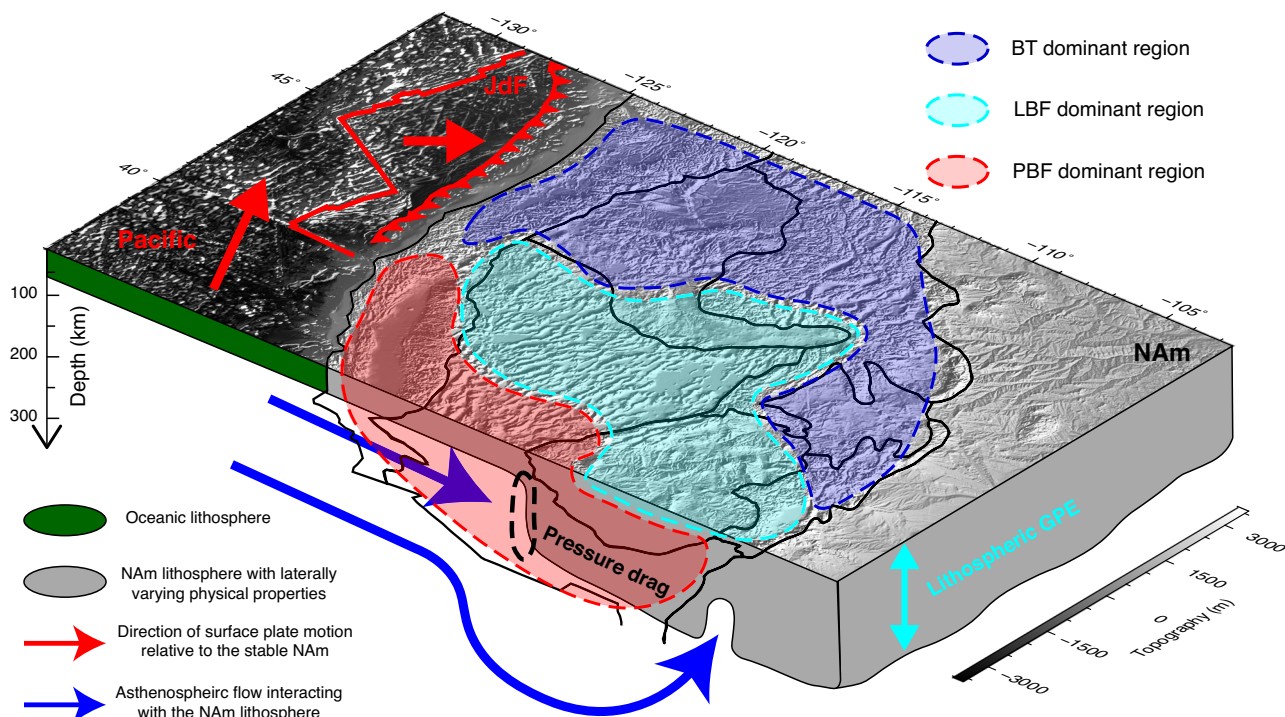

**Fig. 9 | Schematics of the primary driving forces and their relative importance in driving the WUS crustal deformation.** The thin black lines indicate the major tectonic provinces on the surface. The dashed black ellipse outlines the lithospheric thickness step, where the convecting mantle applies pressure drag to the North American lithosphere. BT basal tractions, LBF lithospheric body forces, PBF plate boundary forces, NAm North American plate, JdF Juan de Fuca plate.

The 3D effective lithospheric viscosity structure is critical in modeling lithospheric dynamics[3,4,6,7,14,31]. In our lithosphere model, we use a modified version of the 3D effective lithospheric viscosity structure from Cao and Liu[14]. The continental-scale lithospheric viscosity structure was determined by a parameter search[14]. In the parameter search, the effective viscosity of the WUS, central US, and eastern US were independently varied. The best-fitting lithospheric effective viscosity structure minimized the angular misfit of maximum horizontal compression ($S_{Hmax}$) directions to the observation[25,26] over the conterminous US.

We further infer the fine-scale lithospheric effective viscosity structure in the WUS based on recent seismic attenuation studies[50,51]. Seismic attenuation represents anelastic properties of rocks[79,80] and could be linked to the steady state viscosity by linear viscoelasticity theory. The same set of state variables controls both effective viscosity and seismic attenuation[81,82]. Generally, the seismic attenuation map determines the spatial pattern of effective viscosity but leaves the absolute value undetermined. Therefore, it is possible to infer the effective viscosity structure from seismic attenuation follows the same procedure as in Liu and Hasterok[31]. Since the seismic attenuation in the lithospheric mantle beneath the conterminous US shows regional-scale variations[51], largely consistent with the effective viscosity variations in Cao and Liu[14], we keep the effective viscosity in the lithospheric mantle unchanged. The fine-scale 3D crustal viscosity structure is converted from a recent crustal seismic attenuation study[50]. The crustal seismic attenuation shows the integrated effect through the whole crust. Practically, to construct the 3D effective viscosity structure, we first assume a linear crustal geotherm with lateral varying crustal thickness within the WUS. Then, we determine the temperature-dependent effective viscosity as a reference viscosity structure. Finally, we use the crustal attenuation map to introduce lateral variations at all depths, assuming seismic attenuation is evenly distributed in the radial direction. The whole process could be described by the following equations:

$$\eta = \beta * \eta_0 * \exp\left(\frac{E_c}{T + T_0} - \frac{E_c}{T_c + T_0}\right) \qquad (1)$$

$$\beta = a * \left(\frac{Q}{Q_0}\right)^c \qquad (2)$$

where η is the effective viscosity, β is a coefficient controlling the magnitude of lateral variations, $\eta_0$ is the reference viscosity, $E_c$ is the non-dimensional activation energy, $T$ is the non-dimensional temperature, $T_0$ is the surface temperature, $T_c$ is the characteristic crustal temperature, $Q$ is the crustal attenuation, $Q_0$ is the reference crustal attenuation, $a$ and $c$ are numerical coefficients. $T_c$, $\eta_0$, and $Q_0$ are fixed in our parameter search. An extensive parameter search is performed to find the best-fitting conversion coefficients that minimize the averaged angular misfit to the observed $S_{Hmax}$ directions[25,26] (Supplementary Fig. 7). Through the test, we find the best-fitting coefficients are $a = 10$, $E_c = 3$, and $c = 3$.

We want to emphasize that this approach only approximates the real 3D effective viscosity structure. A better way is to use the 3D magnetotelluric (MT) data inferred effective viscosity structure, which is critical in understanding lithospheric dynamics[31,83,84]. However, due to currently limited MT data coverage, it does not allow a survey covering the whole conterminous US. In the WUS, our effective viscosity model captures the main features revealed by a most recent MT survey[85].

## Calculating crustal stress and deformation

Our numerical models output the full stress tensor and the second invariant of the strain rate tensor on every computing node. We

radially integrate the full stress tensor and the second invariant of the strain rate tensor down to 30 km (10 km for Supplementary Fig. 3). Subsequently, we calculate the depth-averaged horizontal principal stress and the crustal deformation rate (i.e., the second invariant of the strain rate tensor).

The predicted surface motion is output in a lower mantle fixed frame. We take the direct output and remove the North American plate motion[63] relative to the fixed lower mantle to get the relative crustal motion.

### Evaluating predicted mantle flow against seismic observation
Our convecting mantle structure is adopted from a time-dependent mantle flow model[44], which matches multiple geophysical and geological observations[36,37,46]. After incorporating a geophysically constrained lithosphere with lateral varying thickness, we further evaluate the predicted mantle flow by comparing the active mantle flow to the recent observation of depth-dependent seismic azimuthal anisotropy[52], which is not included in our previous studies. We define the active mantle flow as the asthenospheric material having a velocity larger than 0.01 cm/yr in the North American plate fixed reference frame[16]. Subsequently, we calculate the angular misfit between the predicted active mantle flow field and observed seismic azimuthal anisotropy[52].

### Seismic prediction power analysis
We perform a seismic prediction power analysis to compare the predicted crustal deformation rate field and observed seismicity. First, we calculate the normalized deformation rate field, which keeps the spatial pattern, by normalizing the prediction with the maximum predicted value within the WUS. Then, we sample the normalized crustal deformation rate at 26,577 earthquake epicenters during 1980–2020 (Fig. 1a). The quantitative relation between crustal deformation rate and seismicity remains unknown. Nevertheless, recent geodetic studies showed that more than 90% of seismicity happened in areas with geodetically measured crustal strain rate between $2 \times 10^{-9}$ to $2 \times 10^{-7}$ yr$^{-1}$(ref. 59). Therefore, we assume the model successfully predicts an earthquake if the normalized crustal deformation rate is larger than 0.01, within two orders of magnitude from the maximum, at the epicenter. Finally, we calculate the success percentage to quantify the match between the predicted crustal deformation rate field and observed seismicity.

### Evaluating predicted crustal stress and surface motion
To quantify the math between the predicted crustal stress field and the observed crustal stress patterns, we calculate the $S_{Hmax}$ directions and compare them to the observation[25,26] following the same method as Cao and Liu[14]. We also calculate the $A\varphi$ parameter flowing ref. 86 to determine the faulting regime predicted by the model. Since the $A\varphi$ parameter is less well-constrained than $S_{Hmax}$ directions[26], we only qualitatively compare the prediction and observation (Fig. 7c).

The predicted crustal motion is compared to a smoothed version of GPS measurements (spline interpolated from the raw data) on a 0.5° × 0.5° grid over the WUS (black arrows in Figs. 4d–f and 5d–f). The residual velocity, the vectorial difference between the prediction and observation, is calculated on each node, and its average magnitude and average angular misfit are used to quantify the match between the prediction and observation.

### Effects of 3D crustal effective viscosity structure
We perform four additional models (M7–M10) with different lithospheric and convective mantle structures to quantify how the 3D crustal effective viscosity structure affects the contribution of mantle convection to crustal deformation. With a uniform lithosphere (M7), consisting of LTM2 and constant lithospheric effective viscosity, the predicted crustal deformation shows diffusive patterns and decay

monotonically with the distance away from the plate boundaries (Supplementary Fig. 5a), leading to a seismicity prediction power of 73.10%. Within the regions far from the plate boundaries, such as the northern RM and WP, negligible deformation occurs. Besides, the predicted surface motion shows significant northwestward motion in the southwestern US and northward to northeastward motion in the PNW (Supplementary Fig. 5e), with a large average angular misfit (46.7°) and a significant average residual (4.50 mm/yr). The predicted clockwise rotation generally reflects the strong coupling with the fast-moving Pacific plate and the subducting JdF plate. Most of this clockwise motion is solid-body rotation, which does not involve internal deformation of the plate (cf. Supplementary Fig. 5a, e).

With the inclusion of basal tractions (M9), the predicted crustal deformation still maintains a smooth spatial pattern (Supplementary Fig. 5c). The seismicity prediction power increases to 78.11%. The subducting JdF slab locally enhances basal tractions along the western boundary of SRP, resulting in localized increments in crustal deformation (cf. Supplementary Fig. 5a, c). Overall, the active mantle flow drags the WUS eastwards and induces eastward surface motion, particularly in the PNW, SRP, and RM (cf. Supplementary Fig. 5e, g, magenta arrows). The active mantle flow reduces the angular misfit between predicted and observed surface motion to 34.8°, while the average residual remains similar (4.13 mm/yr in M9).

Subsequently, we introduce a 3D crustal effective viscosity structure into the model and assess its impact on the plate boundary effect (M8). The inclusion of this structure notably alters the predicted crustal deformation field in regions away from plate boundaries (cf. Supplementary Fig. 5a, b), increasing the seismicity prediction power to 88.89%. The predicted crustal deformation shows a more localized pattern with high deformation rates in the WL and at the edge of CP (Supplementary Fig. 5b). Importantly, the 3D crustal effective viscosity structure significantly reduces the coupling between the North American plate and the fast-moving Pacific plate, as evidenced by the reduced clockwise rotation in the predicted surface motion (Supplementary Fig. 5f). This decoupling effect reduces the average residual to 2.53 mm/yr, while the average angular misfit (43.4°) remains similar to M7.

When further adding basal tractions (M10) to M8, the predicted crustal deformation field notably changes in the interior of WUS, particularly in the SRP and northern RM (cf. Supplementary Fig. 5b, d). The increased crustal deformation rate in the interior of WUS corresponds to a seismicity prediction power of 96.94%. The predicted crustal motion generally exhibits more eastward motion compared to M8 (cf. Supplementary Fig. 5f, h, magenta arrows) due to the eastward drag exerted by the mantle flow, with a similar average residual (2.53 mm/yr) and a reduced average angular misfit (34.0°). When compared to M9, the inclusion of the 3D crustal structure (M10) significantly reduces the magnitude of surface velocity residual while leaving the average angular misfit unchanged.

### Effects of lithosphere–asthenosphere interaction on crustal rheology
One notable rheological feature in the geophysically inferred 3D lithospheric effective viscosity structure is a long crustal weak zone along the lithospheric thickness step (Fig. 2g). This weak zone is important in reproducing the fast-deforming ISB (Fig. 5 and Supplementary Fig. 5). Its effective viscosity is independently inferred and is likely controlled by the long-term thermal, compositional, and mechanical evolution on crustal rheology.

Some studies suggested that this weak zone is a result of local volcanism since the Cenozoic due to the intruding hot material[6] or lithospheric delamination[87]. However, a recent geodynamic simulation indicated that there has been prominent hot asthenospheric material intruding from the oceanic mantle beneath the whole WUS since ~12 Ma[36]. This broad intrusion would have heated up the WUS

lithosphere uniformly rather than generating localized weak zones. While the elevated lithospheric GPE resulting from high plateaus may assist the crustal deformation and weakening around the CP[6,8], the reconstructed high Nevadaplano[2] is much broader than the narrow weak zone. This situation is similar to the current WUS, where lithospheric GPE gradients drive broad crustal deformation[7,9] (cf. Fig. 5a, c). Therefore, we propose that the long-term strain localization caused by lithosphere–asthenosphere interaction along the lithospheric thickness step represents a more plausible mechanism for the formation of localized intraplate weak zones and the subsequent development of intraplate seismic belts.

### Formation of the lithospheric thickness step

The formation of the lithospheric thickness step likely depends on the contrasting physical properties of the cratonic and non-cratonic lithosphere, as well as the ancient tectonic processes in the WUS, such as lithospheric delamination[87] and flat subduction[88]. Further research involving time-dependent geodynamic simulations that concurrently compute lithospheric and convective mantle dynamics in a unified physical frame is necessary to investigate how the lithospheric thickness step formed and influenced long-term tectonic processes in the WUS.

## Data availability

The GPS-measured surface motion used in this study was downloaded from GAGE/UNAVCO's website (https://www.unavco.org/data/data.html) in January 2020, and this dataset was produced using the methods described in ref. 89. The earthquake information was downloaded from USGS's database (https://www.usgs.gov/programs/earthquake-hazards/earthquakes). The LITHO 1.0 model is available at https://igppweb.ucsd.edu/~gabi/litho1.0.html, and the CRUST 1.0 model is available at https://igppweb.ucsd.edu/~gabi/crust1.html. The earthquake focal mechanism solutions are available in their original publications, and additional data can be found at https://www.globalcmt.org. All other data used in this study, including asthenospheric seismic anisotropy, seismic tomography models, seismic attenuation data, and geophysical estimations of $S_{Hmax}$, can be accessed from the in-text citations, and additional data can be found at https://ds.iris.edu/ds/products/emc-earthmodels/. The models' inputs and predictions are available at https://doi.org/10.5281/zenodo.10850263.

## Code availability

The computational code CitcomS is freely available at www.geodynamics.org.

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

## Acknowledgements

This work is supported by the Strategy Priority Research Program (Category B) of the Chinese Academy of Sciences XDB0710000 and NSFC grant 92355302 to L.L. We acknowledge the use of the Generic Mapping Tool (GMT) for plotting figures. We are grateful to Liang Liu and Yanchong Li for fruitful discussions.

## Author contributions

L.L. initiated and organized the project. Z.C. prepared the code and performed the numerical simulations. All authors contributed to the data analysis and writing the manuscript.

## Competing interests

The authors declare no competing interests.
