## [Peer Review File · Nature Communications]

Editorial Note: Figure on page 34 of this Peer Review File have been redacted as indicated to remove third-party material where no permission to publish could be obtained.

REVIEWER COMMENTS

Reviewer #1 (Remarks to the Author):

Summary

The manuscript argues that the complicated lithospheric structures of the WUS are the primary cause of the region's widespread and intense intraplate deformation and uses extensive geodynamic simulations to support the argument. The authors first present a model with realistic lithospheric structures capable of producing crustal deformations that match the observations significantly better than models with lower degrees of complexity. They then perform experiments by varying each feature in the model independently to test its effect on the predicted crustal deformations. They finally discuss the possible connections between recent volcanisms in the WUS and the deformations caused by the heterogeneous lithospheric structures in their models.

Recommendation

I personally find the results of the paper interesting and are potentially suitable for publishing on Nature Communications because they may provide additional insights into the mechanisms of the WUS's complicated intraplate deformations, an outstanding question in solid-earth geosciences. Nonetheless, in my opinion, the paper in its current form is poorly organized, has major issues with its input parameters, and contains numerous typos and grammatical errors, which call for an overhaul of the manuscript. I thus recommend a major revision and am happy to review it again after the resubmission. Please see my detailed reviews below.

Major issues

The most outstanding issue with the paper is its organization. The contents of the sections "Reproducing Lithospheric and Mantle Dynamics" and "Effects of the Heterogeneous Lithosphere on Crustal Deformation", two main sections presenting their results, are similar and their relation unclear. The section "Effects of the Heterogeneous Lithosphere on Crustal Deformation" lacks a clear focus and contains too many details of too many models, making it very tedious and difficult to read. I suggest the authors restructure the

two sections to present fewer models in a more organized fashion. In the first section, the authors should show the results of their preferred model, which presumably contains all the realistic heterogeneities in the lithosphere, and compare its crustal-deformation predictions with those from a minimalistic model with only the plate boundary constraints and the convective mantle to establish the importance of accounting for complexities in the lithosphere (The current first section presents three models, whose relations are unclear to me). The second section will then describe the parameter test to find the relative importance of all the complexities. Given this paper is about complexities in the lithosphere, I feel that the authors only need to vary the viscosity and density structures of the crust and mantle lithosphere (no need to test the effects of the convective mantle), so they only need to present six models in this section. The authors should also focus on the improvements to the model predictions when complexities are added instead of describing every detail of their results.

I am skeptical about the lithosphere-thickness model the authors use and the way it is presented in the paper. Based on my understanding, the authors use the lithosphere thickness map from LITHO-1.0, a very coarse-resolution global map estimated mainly using surface-wave data. The authors claim that the map shows significant lithosphere thickness variations within the WUS, but the only variation truly within the region is the one around the Colorado Plateau, whereas the rest of the WUS largely has a constant LAB depth of ~65 km, (Fig. 3a). If the authors are to continue using the current LAB depth model, I suggest that they instead describe it as the LAB depth variation between the Colorado Plateau and the rest of the WUS, which is really what it is.

In contrast to the coarse-scale LAB-depth map from LITHO-1.0, recent studies using body-wave scattered phases, particularly Liu and Shearer, (2021), have revealed finer scale LAB-depth variations within the WUS, which correlate well with the regional tectonics. For instance, Liu and Shearer, (2021) showed significantly thinner lithospheres beneath the boundaries of the Basin and Range Province compared with its interior, which likely will cause additional strain localizations if incorporated into the authors' models. I understand that running their models using a different LAB model may be too much of a burden for the authors, but they should at least acknowledge the presence and

potential effects of finer-scale LAB-depth variations in the WUS instead of ignoring the recent significant progresses in seismically characterizing the LAB. In fact, Fig. S3 shows that compared to the predictions of their preferred model, the observed deformation rate shows a significantly greater degree of localization along the east and west boundary of the northern Basin and Range Province, the exact locations where Liu and Shearer, (2021) observed local thinning of the lithosphere. Therefore, using a finerscale LAB-depth model will likely further improve the agreement between their model predictions and the observations.

I am also concerned about the authors' input lithospheric density models. First, the method section does not clearly state if the density in the crust is spatially variable besides the difference between the Snake River Plain and the rest of the WUS. Second, the authors appear to have modified the thicknesses of the crust and lithosphere based on the surface topography by assuming Airy Isostasy without explicitly stating it. I thus suggest that they clarify if Airy Isostasy is indeed assumed. Third, I am concerned if the authors' modified crustal-thickness map is significantly different from the observed one, which is relatively well-constrained by seismological studies. I thus suggest that they show a comparison between their input crustal thickness and the seismologically constrained one in the supplementary material. Lastly, given the importance of the input density models, I suggest that the authors describe how they built them in greater detail in the method section instead of referring to a previous work unless the journal explicitly limits the section's length.

Regarding the discussion on recent volcanisms at the end of the paper, I do not think their results support their claim that a connection exists between the volcanisms and the lithosphere extensions predicted by their model. Figs 10b and c show that most of the Basin and Range Province is in extension and there is no significant difference between its interior and boundaries. Nonetheless, the volcanisms are predominantly located near its boundaries, suggesting that their distribution is controlled by processes not captured by the model, such as the localized lithosphere thinning at the boundaries of the Basin and Range Province shown by Liu and Shearer, (2021). Therefore, I suggest deleting the

discussion on recent volcanisms.

The manuscript also contains numerous typos and grammatical errors, some of which are listed below. Although they generally do not cause difficulty in evaluating the merit of the paper, they do leave me the impression that the authors are uncaredful while preparing the manuscript.

Minor issues

Line 16: The juxtaposition of the terms “lithospheric” and “mantle” is inappropriate as the lithosphere also includes a mantle part. The authors should use “convective mantle” instead of “mantle”. They made this mistake at several other places, e.g., in Line 106.

Line 31: The authors should use the past tense while describing previous works. For example, here, “realize” should be “realized”. This applies to all other places where previous studies are mentioned.

Line 68: What is considered “intraplate volcanism” by the authors? Specifically, are the Cascadia arc volcanisms included? Fig. 1b seems to suggest that they are. Please clarify.

Line 134: The term “Mesozoic Farallon slab” is confusing. My understanding of the modeling process is that they only compute a present-day snapshot of the mantle-flow field instead of running the model from the Mesozoic. Does it mean the Mesozoic Farallon plate that has since been subducted to its current location beneath the WUS? If so, I would call it the “subducted Farallon slab” to avoid confusion.

Lines 141–144: How does the model “reproduces the late-Cenozoic volcanic history and topographic changes”? How do they predict the SKS splitting measurements from the mantle flow and density anomalies?

Line 184: Reference 47 is not about seismic attenuation and thus probably a typo.

Line 215: The sentence should be changed to “...which may have led to an underestimation of ...”

Line 219: “earthquake frequency” should be changed to “seismicity” as the latter is more commonly used in the seismology community. Besides, why does the second invariant of strain rate reflect the deformation rate?

Lines 222–224: In Fig. 4a, why does the area around the Wyoming craton have a very

low deformation rate?

Line 238: The description of M2 does not seem to match that in Table S1 because here, M2 is described to have a constant density, whereas in Table S1 it has “the preferred density structure”. This is related to my aforementioned confusion about whether the crustal density is allowed to vary laterally.

Line 264: The authors mention changing the “definition of lithospheric GPV” multiple times in the manuscript, but how can they change the definition of a physical quantity?

Line 274: The word “linearly” is too strong. Use “monotonically” instead.

Lines 279–281: Can the authors explain why the subduction of the Juan de Fuca plate causes northward motion of the Pacific Northwest in M4, which seems to be roughly perpendicular to the subduction direction?

Lines 317–318: I do not see a significantly increased strain localization along the lithosphere step in Fig. 5c compared to 5b.

Line 320: What “to the east of the lithosphere step” means is unclear. I guess it really means to the east of the boundary between the Basin and Range Province and Colorado Plateau. See my previous criticism about the authors’ overstatement about the lithosphere-thickness variation in their model.

Lines 330–332: According to Table S1, M7, M8, and M9 have the same structure in the convective mantle but different lithospheric viscosity structures, which is opposite to what this sentence states.

Line 336: In which areas does the subducting plate increase crustal deformation?

Line 443–445: It is impossible to visually determine if the predicted stress field is consistent with the observed focal mechanisms from Fig. 9a. I suggest either removing Fig. 9 and the associated discussions or making a new figure showing the predicted stress field and the stress field estimated from the focal mechanisms using stress inversion.

Besides, Fig. 3 does not have Panel c or e, so it’s probably a typo.

Line 468: Fig. 3 does not have Panel c.

Lines 506–508: This statement is too strong. Fig. 5 and 6 show that the crustal-viscosity variation is at least as important as the lithosphere-thickness variation.

Line 512–514: In fact, among the seismically active regions, only the ISB is located near the LAB step. Furthermore, even the expected high deformation rate of this region can be partially reproduced using models with only the crustal-viscosity heterogeneities (e.g., M8).

Lines 524–526: I don't think the authors can preclude the local-volcanism origin of the crustal weak zone based on the theory cited here because it represents only one possible scenario of hot-material intrusion. Besides, "will heat" should be "would have heated."

Lines 527–531: I don't understand the model proposed here.

Line 585: "allow" should be "allowing".

Lines 589–592: What are the boundary conditions on the northern and southern boundaries of the WUS, which are not bounded by the ocean or the CEUS?

Line 601: "4" should be a superscript.

Lines 608–610: What are the effects of the MLD on the density and viscosity of the mantle lithosphere?

Lines 612–614: This sentence is grammatically incorrect. The authors should either add an "and" after the comma or change the comma to a semicolon.

Line 626: A "the" is missing before "3D".

Lines 636–637: The sentence starting with "Then" is not a complete sentence.

Lines 653–654: A "the" is missing before "3D".

Line 670: A "the" is missing before "vectorial".

Fig. 3: Please add the legends showing what the two sets of arrows represent in (b).

Besides, the caption should emphasize that the azimuthal seismic anisotropy in (a) is observed, not predicted, which confused me for quite a while.

Figs. 4d–f: Add legends.

Figs. 5d–f: Add legends.

Fig. 9: The green dashed lines in (b) can be mistaken as the green bars denoting extensional stresses. Consider change the color.

Please add longitude and latitude labels to all maps.

Table S1: What does "Mantle structure" mean? I assume it means structures in the

convective mantle. Besides, “empty” is inappropriate. Use “none” instead.

Reviewer #2 (Remarks to the Author):

The paper “Western US Intraplate Deformation Controlled by the Complex Lithospheric Structure” by Cao & Liu uses numerical modeling to investigate the contributions of various forces on deformation and the distribution of recent volcanism throughout the US. This paper shows somewhat quantitatively that an underappreciated control on magmatism, deformation, and earthquake distribution in the western US is related to gradients in lithospheric structure.

The authors should be applauded for taking on such an ambitious project, which brings together geodynamic modeling, and geologic and geophysical observations. I think the major conclusions of the authors as stated above are valid, however their reasoning and clarity in communicating how they reached that conclusion is lacking. Much of this is due to the vast amount of models with different parameters that they analyze, which is extremely difficult to follow. This could be fixed by adding a table of M1-M9 with the different conceptual parameterizations (e.g., columns of crustal thickness, LAB thickness, rheology, topography, and rows of “constant” or “variable”). This would make it much easier to follow the comparisons. In addition, comparisons between model fits to data are fundamentally qualitative in how they are currently presented, making it difficult to know if the conclusion reached by the authors is quantitatively supported by their modeling. Another column could be added to the encouraged additional table for misfit to various datasets, like strain, stress orientation, and GPS measurements, even if they only represent relative misfits compared to each other rather than to true values, since many of these are only weakly constrained.

Other issues in clarity exist due to semantic definitions (e.g., L84 “the mantle and lithospheric dynamics”). I assume mantle convection was meant here since mantle can also be lithosphere. Another example is broadly using basal traction to apparently refer to any mantle convective force between the asthenosphere and lithosphere, rather than purely referring to surface-parallel forces). In a framework of understanding contributions of different forces, clearly defining these forces is important. This would also help with the fact that the authors really don’t make any clear conclusions about the relative contributions of body forces, basal tractions, or plate boundary forces, rather simply stating in the discussion that the lithospheric thickness gradients are important. It isn’t clear which force lithospheric thickness gradients should be associated with (it seems that this would affect both body forces and viscous drag, or the authors preferred term, “basal tractions”). Thus, the authors never clearly tie their introduction with their results and discussion.

Regarding the methods, I think there are some unclear points. The authors state that they create a 3D viscosity model “adjusted to match the observed direction patterns of crustal stress”. This seems very circular, as they then go on to say that these viscosity variations are important in controlling crustal

stress orientations, which of course makes sense if they were derived from the stress orientations themselves. I think it is important for the authors to clearly identify what is an input to their model vs what is an output. This is exemplified by the L520 “One notable feature along the lithospheric step is the presence of a long weak zone within the crust”. Here, the authors make it sound like a model result, when it is clearly embedded in the input model.

In conclusion, the takeaway of this paper is important, but the strength of their article is not clear in its current form due to the lack of quantitative comparisons between the different models (and therefore, plate driving forces, which they discuss quite a bit in the Introduction to set up their problem). Also, it is extremely difficult to keep track of all the models and their different assumptions, inputs and results. Lastly, and perhaps most importantly, the lack of clarity in writing and terminology will prevent this manuscript from having the impact that it likely could have given the importance of the result. My recommendation is that a significant amount of manuscript revision needs to be done to make the paper more intelligible, as it is extremely difficult to read and understand in its current form.

Detailed Comments:

Abstract

L21: “Particularly, the interaction between the mantle flow and lithospheric thickness step along the eastern boundary of Basin and Range represents a key driving mechanism for localized intracontinental deformation.” Given that the motivation in the Intro is revealing the contributions of plate boundary forces, lithospheric body forces, and viscous drag to deformation, which category does this fall under? I could see this being a lithospheric body force (normal force exerted on LAB topography from flow) and/or a viscous drag. It’s not clear to me that this study actually illuminates the contributions between the stated body forces. I think the motivation should be changed to clearly apply to the conclusions, or the conclusions should be put in the context of the motivation.

Introduction

L26: uplifting -> uplift

L26-28: The first part of this sentence is grammatically unclear. I assume you mean that geological and geophysical evidence show clear signs of extension.

L33-34: This sentence needs to be linked clearly to the stated 3 categories of driving forces. Alternatively, you could walk back the focus on the 3 categories, and simply state that you are exploring something that is largely unexplored in the context of these numerical models.

L42-44: There has been a lot of research done on the controlling forces of this change in plate motion, and is largely attributed to plate boundary forces. An example paper exploring these forces should be cited (perhaps a geodesy paper).

Fig. 1: I recommend adding the boundaries of the Northern, Central, and Southern Basin and Range provinces since you discuss them later in the paper. You should also cite the reference for LAB depth

L68: Need to clarify that PNW volcanoes are not intraplate. The main edifices are typical calc-alkaline subduction-related volcanoes and shouldn't be included with intraplate volcanoes.

L83-84 and 112: Could you clarify what you mean by "one-way" coupling? Seems like a contradiction. Does "one-way" coupling mean all forces are drag from asth to lithosphere? What effect does "two-way" coupling actually have? It seems that all it implies is that it allows for more spatial complexity in your model.

L84, L97, L105: Recommend changing from "lithospheric and mantle dynamics" to "lithospheric and mantle convective dynamics" or "crustal and mantle dynamics"

L130-L138: There don't seem to be any geological constraints described in this paragraph. All are geophysical

L140: This language makes it very difficult to understand what is a constraint and what is a result. Do you force your model to fit topographic changes? How do you do this as a function of time? What do you mean your model reproduces volcanism?

L149-150: Isn't the slab from seismic tomography, rather than consistent with it? How does it get modeled as segmented?

Fig 3: It would be very helpful to quantify the fit between SKS and predicted flow if you are going to use this as justification for a "better" model. In addition, why should azimuthal anisotropy (an absolute measure of deformation/flow) correlate with predictions relative to the NA reference frame? I know this

comparison is done a lot in geodynamic modeling studies, but it isn't clear to me why it would be valid, as the data and results are in different reference frames

L211-216: This sentence seems out of place and should probably be in the Introduction/Motivation section. It doesn't seem to particularly contribute to the discussion in this paragraph. Also might help with clarity if it is broken into multiple sentences.

L226: dominant-dominated

Fig. 4: Very nice conceptual example of the effects of the different parameterizations. I would recommend this difference be quantified somehow to show the point quantitatively. Also, change caption (b) for clarity: "a lithosphere of constant thickness but varying Moho depth" rather than "same as in M1"

L260: This section title ("Effects of the heterogeneous lithosphere on crustal deformation") is trivially different from the prior section ("Mantle and crustal deformation controlled by the complex lithospheric structure"), neither of which really clearly state the parameters you are going to test in these sections.

Again, clearly defining the differences in the models through a table or some other relatively simple figure would greatly help the clarity of which models are testing which contributions.

L285: May be worth noting briefly how this viscosity is constrained since it seems to have a significant effect (and/or referencing the methods)

L311: Which lithospheric layers are varying? Both crustal thickness and total lithospheric thickness? Also how does this vary from M3? Clearly the results are different but it isn't easy to see how/why it varies from M3

L319: This seems contrary to your results (Fig.5b,c). The model that has variable viscosity but a uniform lithosphere (M5) reproduces very similar predicted strain magnitudes to the varying lithospheric structure (M6). Perhaps quantifying misfit would make your point more clear, but it certainly isn't qualitatively clear that it makes a big difference in stress and strain predictions (although seems to make a difference for GPS predictions)

L326: Should this be a new subsection? Frankly, it isn't clear in this paper when we are transitioning from a Section to a Subsection, and these should be clearly indicated in the section headers

L338-340: Are you talking about only the black arrows here? Should point that out in the reference to the figures (e.g., cf. 5d, 6d, black arrows)

L367: I also find the comparison interesting between M7 and M8. We can clearly see that the magnitude of surface motion is strongly linked to crustal rheology, however the orientations remain nearly identical. However, this comes back to the fact that is glossed over in the methods, where your viscosity field is derived from surface motions. Regardless, this is another place that would benefit from a quantitative analysis rather than purely qualitative comparison

L377-380: This statement is unclear to me. It sounds like the significance of the interaction is that it is not significant given no change in local stress. Maybe change to "variations in the viscosity of the convective mantle clearly plays a secondary role relative to LAB gradients in controlling crustal stresses". Given this, it would imply that viscous drag (or at least basal traction) is a negligible plate driving force in the interior. This would be good to link to the overall discussion on driving forces.

L380-382: May be worth labeling this on the figures? I don't see particularly large changes in defm rate between the two models.

L386-389: I assume these results are for M9? I think it would help to explicitly state somewhere which model these are the predictions for. I assume M9 but it isn't stated. It is hard to keep track of all these models and their differences

Figure 7: State which model this is for: M7? This is also a place where an explicit definition of traction would be helpful. I would assume that (a-c) is the traction parallel to the horizontal plane, but there are clear upward and downward tractions being observed, which I would think should be called normal forces. Again, maybe a semantical difference.

L435: Change "lithospheric thickness and strength" to "lithospheric properties". As you modify crustal thickness as well as lithospheric thickness, the former statement doesn't clearly show all the factors you tested. The "properties" is more vague

L440: "pressure drag" is this a basal traction? or a body force? This isn't made clear

L440: Should this be Fig 7? It isn't stated that lithosphere changes at all in Fig 8.

L447: Should this be Fig 9a?

L460-461: This probably should be cited if you are going to state this. Given the long active tectonic history of the western half of North America, many of these faults are reactivated from prior tectonic events and can rupture even if they are not presently at the optimal orientation relative to the driving forces since they will be weaker than unbroken rock.

L465-468: Many authors attribute the N-S compression to SAF-related tectonic forces as well as JdF plate boundary forces. This should probably be added. How can you decouple plate boundary forces from basal tractions in strike-slip regions?

L470-: A couple things here:

L468-473: How does combining N-S extensional stress with W-E extensional stress lead to W-E extension? This seems to overstate the importance of basal traction given that it doesn't seem to alter GPE predictions. 2) Where are you talking about when you say southern Basin and Range? The southern Basin and Range has a relatively specific location (central to southern Arizona and Mexico; see Jones et al 1992), but there are only 2 or 3 focal mechanisms in this area, only 1 of which seems to be N-S oriented. 3) I would bet that including a model of lithospheric properties in Mexico (such as the subducting slab and LAB variations) would significantly alter the results in the southernmost portion of this model (away from the SAF). This should probably be acknowledged and likely makes the discussion on what factor is controlling stress orientation at these southern latitudes only loosely constrained by measurements in the US

L494-496: But which of the 3 forces is this? You set up a discussion that states that this study contributes to answering which dominates the WUS, but never explicitly link the results to the forces

L508-509: definition of GPE -> do you mean the magnitude of GPE? The definition of GPE doesn't change, it is the values included in the calculation that change.

L509-512: remove also: it seems that the big change between how the lithosphere and asthenosphere act IS through changing the lateral viscosity structure, and thus ease of asthenospheric convection/motion

L516: I recommend using a phrase besides "transition zone" for clarity, like the "transition from thin to thick lithosphere". The transition zone has definitions to the geophysical community (mantle transition zone) as well as the tectonic community in the western US (transition zone is a small tectonic strip between southern B&R extension and the Colorado Plateau). Both of these are pretty aseismic and may just confuse readers

L520: This feature is embedded in your model through your parameterization and the depth to the Moho and LAB. However, you state it here like it is a result of your modeling rather than an input. You should either delete this sentence (the 2nd sentence is the important one for discussion anyway) or modify it to clearly differentiate results from inputs

L521: change "may reflect" to "its rheology is controlled by thermal, compositional, and mechanic properties." Clearly it must be controlled by these, as the weak zone in your crust is embedded through your own parameterization of the model

L549-554: I do not think that the widespread presence of the low velocity zone necessitates non-vertical transport of melt. In your previous paragraph you conclude that (and I agree) that translithospheric extension is likely necessary to get asthenospheric melt to the surface. Given the rather small volume of most (non-Yellowstone related) magmatism in the WUS, why couldn't these localized extension region only tap the asthenosphere below them, while melt away from these zones remains trapped below an impermeable lithosphere due to no vertical strain?

In addition, I don't think it is appropriate to compare mostly small-volume basaltic (i.e., very low viscosity) magmatism that spatially dominates the intraplate WUS to a long-lived rhyolitic (i.e., high viscosity) magmatism, and perhaps even less appropriate to compare locations of short-lived asthenosphericly-sourced magmatism to long-lived magma residence in magma reservoirs. I agree that below more silicic magmatic centers, there is no need for the reservoir to lie below the edifice, as the edifice location is likely controlled by brittle, active structures in the upper crust (especially for high viscosity magmas). There is plenty of literature on that. But I do not think this applies to pathways for very low viscosity melts. Also, you are implying that melt is able to be taps from 100s of km away from where they erupt with this comparison, whereas active magmatic systems generally show reservoirs below edifices, while offset, tend to be no more than ~10 km away from the edifice (Lerner et al., 2020). This mix increase with depth, but it would be quite a stretch to say that the center of the B&R is sourcing magmatism near the Wasatch. Lastly, a simple calculation comparing the volume of the LVZ below the

B&R with some assumed melt percentage with the total erupted volume in the B&R will show that very little assumed melt reaches the surface.

You state a more plausible interpretation in L554-556, by saying melt reaches the surface if lithospheric stresses allow for it. In a more compressional state, the lithosphere could probably be inferred to act as an impermeable cap, even if melt is present.

This could all be corrected by simply deleting the sentence encompassed by L553-554

L560-561: It would be nice to have more specific examples rather than just pointing out large regions. I think the East African Rift system near cratons and the Anatolian system are both areas where literature has interpreted that magmatism correlates with throughgoing lithospheric structures and presumed changes in LAB depth

Methods:

L607: This is a really low resolution model of crustal thickness. Why not use better constrained thicknesses, like from Schmandt et al 2015 (model on Earthscope EMC page)

L627-629: This seems circular...Of course adding 3D viscosity variations to your model in this study leads to better fits to crustal stresses, the viscosity model is derived from fitting the stresses!

Below are our detailed point-to-point responses to the reviewers' comments and suggestions, with original comments in black and our response in blue.

To reviewer #1:

I personally find the results of the paper interesting and are potentially suitable for publishing on Nature Communications because they may provide additional insights into the mechanisms of the WUS's complicated intraplate deformations, an outstanding question in solid-earth geosciences. Nonetheless, in my opinion, the paper in its current form is poorly organized, has major issues with its input parameters, and contains numerous typos and grammatical errors, which call for an overhaul of the manuscript. I thus recommend a major revision and am happy to review it again after the resubmission. Please see my detailed reviews below.

Response: Thanks for your kind words and insightful comments regarding our manuscript. We greatly appreciate your constructive feedback. To enhance the quality of our manuscript, we have made significant revisions based on your comments and suggestions. Specifically, we have reorganized our "Results and Discussion" section to enhance the logical flow and clarity. Additionally, we have expanded the "Methods" section to provide more details about our model inputs, assumptions, and justifications for our modeling approach. Furthermore, we have introduced a discussion on the potential links between our model and recent advancements in seismological studies and deleted all volcanism-related content. Thanks again for your constructive reviews.

The most outstanding issue with the paper is its organization. The contents of the sections "Reproducing Lithospheric and Mantle Dynamics" and "Effects of the Heterogeneous Lithosphere on Crustal Deformation", two main sections presenting their results, are similar and their relation unclear. The section "Effects of the Heterogeneous Lithosphere on Crustal Deformation" lacks a clear focus and contains too many details of too many models, making it very tedious and difficult to read. I suggest the authors restructure the two sections to present fewer models in a more organized fashion. In the first section, the authors should show the results of their preferred model,

which presumably contains all the realistic heterogeneities in the lithosphere, and compare its crustal-deformation predictions with those from a minimalistic model with only the plate boundary constraints and the convective mantle to establish the importance of accounting for complexities in the lithosphere (The current first section presents three models, whose relations are unclear to me). The second section will then describe the parameter test to find the relative importance of all the complexities. Given this paper is about complexities in the lithosphere, I feel that the authors only need to vary the viscosity and density structures of the crust and mantle lithosphere (no need to test the effects of the convective mantle), so they only need to present six models in this section. The authors should also focus on the improvements to the model predictions when complexities are added instead of describing every detail of their results.

Response: Thanks for your valuable suggestions! We have divided the “Results and Discussion” section into several subsections to enhance the logic flow. In the revised “Results and Discussion” section, we first present how we construct the fully dynamic 3D model. Then, we discuss the impact of lateral varying lithospheric thickness on the asthenospheric flow. Following that, we show our results on crustal deformation in two sections. In the first section, we show the necessity of a complex 3D lithospheric structure in reproducing the observed crustal deformation pattern. Then, we show how a heterogeneous lithosphere influences the contribution of mantle convection to crustal deformation through the plate boundary effect and basal tractions and estimate the crustal deformation and stress resulting from lithospheric body forces. After that, we present the results on locally increased lithosphere-asthenosphere interaction beneath the western US (WUS). Finally, we discuss the relative importance of the proposed driving mechanisms accounting for the realistic lithospheric structure.

Previous studies using simplified lithospheric structures suggested that the mantle convection (e.g., Steinberger et al., 2001, EPSL; Becker et al., 2015, Nature) and crustal gravitational potential energy (e.g., Jones et al., 1996, Nature; Artyushkov, 1973, JGR) are two main drivers for crustal deformation. Therefore, we consider these two factors first in the “Complex Lithospheric Structure Controls WUS Crustal Deformation” section. However, the predicted crustal deformation field fails to reproduce the observed crustal deformation field. Then, we show the predicted crustal

deformation from our best-fitting model to highlight the importance of a complex three-dimensional lithosphere.

In the “WUS Crustal Deformation due to Different Driving Mechanisms” section, we have retained all six models because lateral variations in the lithospheric thickness have a profound impact on the interaction between the continental lithosphere and convecting mantle, including the plate boundary effect and basal tractions. A heterogeneous lithosphere will impact how the plate boundary forces transmit into the interior. Also, when there is a flat lithosphere-asthenosphere boundary (LAB), the convecting mantle exerts horizontal shear and normal force in the radial direction along the LAB. However, in regions with a strong lateral gradient in lithospheric thickness, such as along the eastern boundary of the Basin and Range (B&R), the mantle flow exerts normal force in the horizontal direction, as demonstrated in Fig. 7e and 7f. This force was previously overlooked in the studies assuming a flat LAB (e.g., Flesch et al., 2000, *Science*; Becker et al., 2016, *Nature*; Bahadori et al., 2022, *Nature Communications*). Therefore, we have retained all six models to fully explore the importance of a heterogeneous lithosphere on the contribution of mantle convection to crustal deformation. To enhance the clarity in this section, we have revised the text to focus on the predicted crustal deformation patterns and the improvements when complexities are added, according to your suggestions.

I am skeptical about the lithosphere-thickness model the authors use and the way it is presented in the paper. Based on my understanding, the authors use the lithosphere-thickness map from LITHO-1.0, a very coarse-resolution global map estimated mainly using surface-wave data. The authors claim that the map shows significant lithosphere-thickness variations within the WUS, but the only variation truly within the region is the one around the Colorado Plateau, whereas the rest of the WUS largely has a constant LAB depth of ~65 km, (Fig. 3a). If the authors are to continue using the current LAB-depth model, I suggest that they instead describe it as the LAB depth variation between the Colorado Plateau and the rest of the WUS, which is really what it is.

Response: We really appreciate your insight! Our lithospheric thickness model is a hybrid one, incorporating LITHO 1.0 (Payanos et al., 2014, *JGR*) and a regional body-wave tomography model by Schmandt & Lin (2014, *GRL*). LITHO 1.0 characterizes a thin lithosphere with smooth lateral

variations in the WUS, including the Intermountain West (IW) and Colorado Plateau (CP). In contrast, the body-wave tomography revealed a continuous fast anomaly extending beyond 150 km in the IW and CP. To capture the first-order lithospheric thickness variation, we combined LITHO 1.0 with the body-wave tomography model to create a continental-scale lithospheric thickness model. In practice, we adjusted the WUS lithospheric thickness derived from LITHO 1.0 by tracking continuous fast anomalies in the body-wave tomography model. We have added a section in the revised “Methods” describing how we construct the lithospheric thickness model (lines 1738-1745). Also, in the new section, “Construction of the Fully Dynamic 3D Model”, we have specified that the lithospheric thickness variation discussed in the manuscript is the lithospheric thickness step along the eastern boundary of B&R (lines 444-447).

In contrast to the coarse-scale LAB-depth map from LITHO-1.0, recent studies using body-wave scattered phases, particularly Liu and Shearer, (2021), have revealed finer scale LAB-depth variations within the WUS, which correlate well with the regional tectonics. For instance, Liu and Shearer, (2021) showed significantly thinner lithospheres beneath the boundaries of the Basin and Range Province compared with its interior, which likely will cause additional strain localizations if incorporated into the authors’ models. I understand that running their models using a different LAB model may be too much of a burden for the authors, but they should at need acknowledge the presence and potential effects of finer-scale LAB-depth variations in the WUS instead of ignoring the recent significant progresses in seismically characterizing the LAB. In fact, Fig. S3 shows that compared to the predictions of their preferred model, the observed deformation rate shows a significantly greater degree of localization along the east and west boundary of the northern Basin and Range Province, the exact locations where Liu and Shearer, (2021) observed local thinning of the lithosphere. Therefore, using a finer-scale LAB-depth model will likely further improve the agreement between their model predictions and the observations.

Response: Thanks for directing us to recent seismological studies. Using a finer-scale lithospheric thickness model may further improve the agreement between our model prediction and the observations. However, we want to highlight the first-order importance of lithospheric thickness variation on lithospheric deformation and shallow mantle flow, as they were neglected in most previous geodynamic studies. Thus, a regional-scale lithospheric thickness model, including the

first-order lithospheric thickness contrast between the actively deforming WUS and the rest of the conterminous US, is enough for this purpose. To acknowledge the recent advances in seismological studies, we have introduced a paragraph discussing our model results and the observed lithospheric thinning in the revised manuscript (lines 1254-1262).

I am also concerned about the authors' input lithospheric density models. First, the method section does not clearly state if the density in the crust is spatially variable besides the difference between the Snake River Plain and the rest of the WUS. Second, the authors appear to have modified the thicknesses of the crust and lithosphere based on the surface topography by assuming Airy Isostasy without explicitly stating it. I thus suggest that they clarify if Airy Isostasy is indeed assumed. Third, I am concerned if the authors' modified crustal-thickness map is significantly different from the observed one, which is relatively well-constrained by seismological studies. I thus suggest that they show a comparison between their input crustal thickness and the seismologically constrained one in the supplementary material. Lastly, given the importance of the input density models, I suggest that the authors describe how they built them in greater detail in the method section instead of referring to a previous work unless the journal explicitly limits the section's length.

Response: Thanks for the comments! We have added a detailed description of our lithospheric density structure in the revised "Methods" section (lines 1774-1787). In our preferred lithospheric density structure, we do not allow lateral density variations in the crust, except for the Snake River Plain (SRP) due to its enrichment in basaltic composition (DeNosaquo et al., 2009, JVGR; McCurry & Rodgers, 2009, JVGR).

During the modification of lithospheric and crustal thickness, we did not assume Airy Isostasy. Practically, we adjusted the LAB and Moho in two steps, and we ran a simulation with all density anomalies in the continental lithosphere and convective mantle in each step to get the predicted topography, which contains both dynamic and isostatic components. Based on the difference between predicted topography and observed topography, we first adjusted the LAB depth due to its larger uncertainty compared to the Moho depth. Then, we did another iteration to adjust the Moho depth. The resulting Moho depth is consistent with recent seismological studies. We have also included a figure in the supplementary material (Supplementary Fig. 5) showing the

comparison between our input Moho model and recent seismologically imaged Moho in the WUS (Laske et al., 2013; Schmandt et al., 2015, GRL).

Regarding the discussion on recent volcanisms at the end of the paper, I do not think their results support their claim that a connection exists between the volcanisms and the lithosphere extensions predicted by their model. Figs 10b and c show that most of the Basin and Range Province is in extension and there is no significant difference between its interior and boundaries. Nonetheless, the volcanisms are predominantly located near its boundaries, suggesting that their distribution is controlled by processes not captured by the model, such as the localized lithosphere thinning at the boundaries of the Basin and Range Province shown by Liu and Shearer, (2021). Therefore, I suggest deleting the discussion on recent volcanisms.

Response: Thank you for this comment, which we agree with. In the revised manuscript, we have deleted the discussion on recent intraplate volcanism according to your suggestion.

The manuscript also contains numerous typos and grammatical errors, some of which are listed below. Although they generally do not cause difficulty in evaluating the merit of the paper, they do leave me the impression that the authors are uncaredful while preparing the manuscript.

Response: We really appreciate this comment. During the revision, we carefully went over the grammar and checked all spells, hoping the revision is grammatically correct and logically sound.

Minor issues

Line 16: The juxtaposition of the terms “lithospheric” and “mantle” is inappropriate as the lithosphere also includes a mantle part. The authors should use “convective mantle” instead of “mantle”. They made this mistake at several other places, e.g., in Line 106.

Response: Thank you for pointing this out! We have changed the expression to “lithosphere and convective mantle”. In addition, we have applied this change to all related places.

Line 31: The authors should use the past tense while describing previous works. For example, here, “realize” should be “realized”. This applies to all other places where previous studies are mentioned.

Response: We really appreciate this suggestion. During the revision, we have changed all places describing previous studies to the past tense.

Line 68: What is considered “intraplate volcanism” by the authors? Specifically, are the Cascadia arc volcanisms included? Fig. 1b seems to suggest that they are. Please clarify.

Response: Here, we want to use the term “intraplate volcanism” to describe the volcanoes away from active subduction zones. The volcanoes in the Cascadia arc should not be included in this case. In the revision, we have removed all volcanism-related content from the manuscript.

Line 134: The term “Mesozoic Farallon slab” is confusing. My understanding of the modeling process is that they only compute a present-day snapshot of the mantle-flow field instead of running the model from the Mesozoic. Does it mean the Mesozoic Farallon plate that has since been subducted to its current location beneath the WUS? If so, I would call it the “subducted Farallon slab” to avoid confusion.

Response: This is a good point. Our present-day convective mantle structure is taken from a time-dependent mantle flow model starting from the Late Cenozoic (20 Ma, Zhou & Liu, 2017, G-cubed). The term “Mesozoic Farallon slab” means the Farallon Plate has been subducted beneath North America since the Mesozoic. As recommended, we have changed it to “subducted Farallon slab” to avoid ambiguity (line 412).

Lines 141–144: How does the model “reproduces the late-Cenozoic volcanic history and topographic changes”? How do they predict the SKS splitting measurements from the mantle flow and density anomalies?

Response: These are good questions! As stated above, our convective mantle structure is taken from a time-dependent mantle flow model starting from the Late Cenozoic (20 Ma, Zhou & Liu,

2017, G-cubed). The predicted time-dependent mantle flow is used to investigate the spatial correlation between the intruding hot asthenospheric material and intraplate volcanism (Zhou et al., 2018, Nature Geoscience), calculate flow-induced asthenospheric anisotropy and SKS splitting (Zhou et al., 2018, EPSL), and quantify the change in dynamic topography (Zhou & Liu, 2019, EPSL). These studies successfully reproduced the late Cenozoic WUS volcanic history, present-day observed SKS splitting measurements in the WUS, and WUS topographic changes since the Miocene. These calculations are outside of this manuscript's scope, so we invite the reviewer and readers to check our previous publications for more details. We have revised this part to clearly state that our present-day convecting mantle structure is adopted from a time-dependent mantle flow model, and the time-dependent convecting mantle structure matches the mentioned geophysical and geological observations, as shown in our previous studies (lines 407-440).

Line 184: Reference 47 is not about seismic attenuation and thus probably a typo.

Response: Yes, this is a typo. The right reference should be 48 in the original order (50 in the revised manuscript), we have corrected this typo.

Line 215: The sentence should be changed to "...which may have led to an underestimation of..."

Response: Thank you for this suggestion. We have substantially reorganized and revised this section, and this sentence has been removed from the revised manuscript.

Line 219: "earthquake frequency" should be changed to "seismicity" as the latter is more commonly used in the seismology community. Besides, why does the second invariant of strain rate reflect the deformation rate?

Response: We have changed the term "earthquake frequency" to "seismicity" in this sentence and all related parts accordingly.

Generally, the second invariant of a tensor is a scalar measurement of its magnitude. Specifically for the strain rate tensor, its second invariant measures the total magnitude of the deformation rate,

including dilatation (volume change) and shear. This is a widely used scalar quantity of deformation rate in the geodetic community (e.g., Kreemer et al., 2003, G-cubed), and it has been shown to be a good proxy for seismicity (e.g., Bird & Kreemer, 2015, BSSA; Kreemer et al., 2022, SRL).

Lines 222–224: In Fig. 4a, why does the area around the Wyoming craton have a very low deformation rate?

Response: This is a good observation! In this model, the crustal deformation pattern is dominated by the plate boundary effect, which monotonically decays with distance. The Wyoming plateau (WP) is situated far from the plate boundaries, so the deformation rate is low within the WC and its surrounding area.

Line 238: The description of M2 does not seem to match that in Table S1 because here, M2 is described to have a constant density, whereas in Table S1 it has “the preferred density structure”. This is related to my aforementioned confusion about whether the crustal density is allowed to vary laterally.

Response: M2 has the preferred crustal density structure, in which the crustal density is constant (2850 kg/m^3), except for the SRP. Since the crust in the SRP is enriched in basaltic composition, we assume the average crustal density in the SRP is denser (2950 kg/m^3) than in other places. This assumption is consistent with geological and geophysical studies (e.g., DeNosaquo et al., 2009, JVGR; McCurry & Rodgers, 2009, JVGR; Zhou & Liu, 2019, EPSL). We have modified the description here (lines 757-760) and added details about our crustal density structure in the revised Methods session to avoid ambiguity (lines 1774-1787).

Line 264: The authors mention changing the “definition of lithospheric GPE” multiple times in the manuscript, but how can they change the definition of a physical quantity?

Response: This is a great point! Traditionally, the lithospheric gravitational potential energy (GPE) is calculated under the assumption of the existence of a global compensation depth such as at 100 km (e.g., Ghosh et al., 2012; Becker et al., 2016, Nature; Bahadori et al., 2019, Geosphere).

However, the assumed global compensation depth is questionable and may vary geographically (Schubert & Turcot, 2002). Here, we mean the traditional calculation with this implicit assumption by using the term “definition of lithospheric GPE”. However, the word ‘definition’ is indeed a bit confusing. In the revised manuscript, we have changed this term to ‘calculation’, and added a section to describe how we calculate three types of driving forces and discuss the differences between our calculations and those in previous studies (lines 803-835). Particularly, we pointed out the problem embedded in the conventional calculation of lithospheric GPE (lines 827-829).

Line 274: The word “linearly” is too strong. Use “monotonically” instead.

Response: We have changed the word “linearly” to “monotonically”. Thanks for the suggestion.

Lines 279–281: Can the authors explain why the subduction of the Juan de Fuca plate causes northward motion of the Pacific Northwest in M4, which seems to be roughly perpendicular to the subduction direction?

Response: This is a good observation! In M4, the general northwestward to northward surface motion is dominated by the drag from the fast-moving Pacific plate due to the strong coupling. The subducting Juan de Fuca (JdF) plate causes northeastward surface motion in the subduction direction. The combination of the northwestward motion, caused by the right-lateral shear along the Pacific-North America plate, and the northeastward motion, caused by the subducting JdF plate, results in an apparent northward surface motion in the Pacific Northwest.

Lines 317–318: I do not see a significantly increased strain localization along the lithosphere step in Fig. 5c compared to 5b.

Response: Here, we want to emphasize that the deformation is concentrated along the lithospheric thickness step on the thin and weak side (i.e., the eastern boundary of B&R) in M6 (Fig. 5c). In comparison, the crustal deformation diffuses into the interior of CP and WP in M5 (Fig. 5b). We have modified this part to make our point more straightforward by deleting the statement about more localization. Besides, we have added a description clearly pointing to the change in the interior of CP and WP (lines 978-980).

Line 320: What “to the east of the lithosphere step” means is unclear. I guess it really means to the east of the boundary between the Basin and Range Province and Colorado Plateau. See my previous criticism about the authors’ overstatement about the lithosphere-thickness variation in their model.

Response: Here, “to the east of the lithospheric step” means to the east of the lithospheric thickness step along the eastern boundary of B&R (the orange dashed line in Fig. 5c). We have added a description in lines 444-447 to specify that the lithospheric thickness variations discussed in this manuscript is the sharp lithospheric thickness step along the eastern boundary of B&R.

Lines 330–332: According to Table S1, M7, M8, and M9 have the same structure in the convective mantle but different lithospheric viscosity structures, which is opposite to what this sentence states.

Response: This is a good point. Here, we extracted basal tractions by calculating the difference between two predicted stress fields with the same lithospheric structure but different convective mantle structures. For example, the stress field shown in Fig. 6c (now Fig. 5f) is the difference between M9 and M6, which have the same lithospheric structure but different convective mantle structures (M9 has all density anomalies in the convective mantle, but M6 has an empty mantle). We have rewritten this sentence to improve its clarity (lines 866-868) and revised the figure to only show the estimated crustal stress field due to different mechanisms when the realistic 3D lithosphere is included in the model.

Line 336: In which areas does the subducting plate increase crustal deformation?

Response: In the PNW back arc region, i.e., near the western boundary of the SRP. We have added a more specific description in this sentence (lines 899-901).

Line 443–445: It is impossible to visually determine if the predicted stress field is consistent with the observed focal mechanisms from Fig. 9a. I suggest either removing Fig. 9 and the associated discussions or making a new figure showing the predicted stress field and the stress field estimated

from the focal mechanisms using stress inversion. Besides, Fig. 3 does not have Panel c or e, so it's probably a typo.

Response: Thanks for the suggestion! We have revised Fig. 9 to include two panels comparing our model predictions with recent geophysical estimations of S_{Hmax} direction and $A\phi$ parameter (Lund Snee & Zoback, 2020, Nature Communications). Here, Fig. 3 should be Fig. 4. We have corrected this typo.

Line 468: Fig. 3 does not have Panel c.

Response: Same as the above, it should be Fig. 4. We have corrected it.

Lines 506–508: This statement is too strong. Fig. 5 and 6 show that the crustal-viscosity variation is at least as important as the lithosphere-thickness variation.

Response: We really appreciate this comment! The 3D crustal effective viscosity is important in reproducing the observed crustal deformation pattern. However, we analyzed the potential mechanisms for creating the localized crustal weak zone and found that the long-term localized lithosphere-asthenosphere interaction may ultimately control it. Therefore, we think this statement is generally valid. We have brought up the discussion on formation mechanisms for crustal weak zones and merged it with our results showing localized lithosphere-asthenosphere interaction into a new section, “Mechanism for Localized Intraplate Deformation” (lines 1187-1262).

Line 512–514: In fact, among the seismically active regions, only the ISB is located near the LAB step. Furthermore, even the expected high deformation rate of this region can be partially reproduced using models with only the crustal-viscosity heterogeneities (e.g., M8).

Response: Here, we refer to “intraplate earthquakes” as the seismicity in the ISB. The crustal rheology likely reflects the long-term thermal, compositional, and mechanical effects of tectonic events. By analyzing the potential driving mechanisms, we propose that the crustal weak zone in the ISB was created by the long-term strain localization caused by lithosphere-asthenosphere interaction along the lithospheric thickness step (lines 1222-1252). Therefore, the varying

asthenospheric flow ultimately controls the seismicity in the ISB. In the revised manuscript, this sentence has been removed.

Lines 524–526: I don't think the authors can preclude the local-volcanism origin of the crustal weak zone based on the theory cited here because it represents only one possible scenario of hot-material intrusion. Besides, "will heat" should be "would have heated."

Response: This is a good point. First, the hot material intrusion predicted by Zhou et al. (2018, Nature Geoscience) is consistent with the surface volcanism trend since 16 Ma. This broad intrusion has been covering the entire upper-most asthenosphere beneath the WUS since ~12 Ma (Fig. 4 in Zhou et al., 2018, Nature Geoscience), aligning with the onset of volcanism along the eastern boundary of B&R. Besides, the same time-dependent mantle flow predicted flow-induced seismic anisotropy (Zhou et al., 2018, EPSL) and dynamic topography change since the Miocene (Zhou & Liu, 2019, EPSL), consistent with observations over the entire WUS. Therefore, this scenario seems to be the most plausible one.

In addition, the localized crustal weak zone extends from the southern B&R all the way to the northern Rocky Mountain (RM), but the Cenozoic volcanism only clustered around the edge of CP and in the SRP. Notably, the crust has low effective viscosity in the Wasatch Fault Zone and northern RM, where no Cenozoic intraplate volcanoes exist. The disagreement between the spatial pattern of the crustal weak zone and intraplate volcanism indicates the volcanic origin is not valid. In contrast, since ~12 Ma, the eastward hot asthenospheric flow has been continuously interacting with the lithospheric thickness step (Zhou et al., 2018, Nature Geoscience), which spatially coincides with the entire crustal weak zone, providing a more plausible mechanism for the formation of the weak zone.

Moreover, the majority of intraplate volcanoes along the eastern boundary of B&R were fueled by small-volume basaltic melt with low viscosity, which has limited ability to channel through the entire lithosphere. Forming these basaltic volcanoes may further require pre-existing lithospheric weak zones, rather than creating the weak zones.

Lines 527–531: I don't understand the model proposed here.

Response: Here, we assess another potential formation mechanism for the narrow crustal weak zone. The high plateau, such as the Nevadaplano, will locally increase the extensional stress due to the elevated crustal GPE (e.g., Jones et al., 1996, Nature). The increased extensional stress may cause intense localized crustal deformation, leading to the formation of a crustal weak zone. However, the reconstructed Nevadaplano is much wider than the weak zone converted from seismic attenuation, and it only covered the boundary between the B&R and CP. Thus, it is hard to explain the formation of the long, narrow crustal weak zone extending all the way to the northern RM by this hypothesis.

Line 585: “allow” should be “allowing”.

Response: We have changed the word accordingly (line 1600).

Lines 589–592: What are the boundary conditions on the northern and southern boundaries of the WUS, which are not bounded by the ocean or the CEUS?

Response: This is a good question! Our model includes the conterminous US and parts of Canada and Mexico. In the model, we apply plate motions in the surrounding ocean basins, stable central and eastern US (east of 110°E) and Canada (north of 55°N). Other places all have a free-slip boundary condition. We have added some details about the hybrid surface velocity boundary conditions in the revised “Methods” section (lines 1605-1608).

Line 601: “4” should be a superscript.

Response: Yes, it is a superscript, and we have corrected it.

Lines 608–610: What are the effects of the MLD on the density and viscosity of the mantle lithosphere?

Response: We take the MLD as a boundary separating the layered continental lithospheric mantle (CLM). In our lithospheric density model, the upper and lower CLM have different compositional densities. The lower CLM is compositionally denser than the upper CLM. This configuration is supported by our recent works (e.g., Zhou et al., 2019, EPSL; Hu et al., 2019, Nature Geoscience; Cao & Liu, 2021, JGR; Wang et al., 2021, GRL; Wang et al., 2021, JGR; Wang et al., 2023, Nature Geoscience). The MLD does not affect the effective viscosity in our model. We have added details regarding the effects of the MLD in lines 1744-1772.

Lines 612–614: This sentence is grammatically incorrect. The authors should either add an “and” after the comma or change the comma to a semicolon.

Response: We have added an “and” after the comma.

Line 626: A “the” is missing before “3D”.

Response: We have added the article before “3D”.

Lines 636–637: The sentence starting with “Then” is not a complete sentence.

Response: We completed this sentence by adding “we” before “determine”.

Lines 653–654: A “the” is missing before “3D”.

Response: We have added the article in the sentence.

Line 670: A “the” is missing before “vectorial”.

Response: We have added the article before “vectorial”.

Fig. 3: Please add the legends showing what the two sets of arrows represent in (b). Besides, the caption should emphasize that the azimuthal seismic anisotropy in (a) is observed, not predicted, which confused me for quite a while.

Response: Thank you for the comment! We have modified the figure to include the suggestions.

Figs. 4d–f: Add legends. Figs. 5d–f: Add legends.

Response: In these figures, we include the velocity scale at the bottom of the figure and specify the meaning of the colors in the figure caption.

Fig. 9: The green dashed lines in (b) can be mistaken as the green bars denoting extensional stresses. Consider change the color.

Response: We have changed the color of the dashed line to orange.

Please add longitude and latitude labels to all maps.

Response: We have added the longitude and latitude labels to all maps.

Table S1: What does “Mantle structure” mean? I assume it means structures in the convective mantle. Besides, “empty” is inappropriate. Use “none” instead.

Response: Yes, it should be the convective mantle structure. We have revised the supplementary table S1 to address this issue and changed “none” to empty in the table.

To reviewer #2:

The authors should be applauded for taking on such an ambitious project, which brings together geodynamic modeling, and geologic and geophysical observations. I think the major conclusions of the authors as stated above are valid, however their reasoning and clarity in communicating how they reached that conclusion is lacking. Much of this is due to the vast amount of models with different parameters that they analyze, which is extremely difficult to follow. This could be fixed by adding a table of M1-M9 with the different conceptual parameterizations (e.g., columns of crustal thickness, LAB thickness, rheology, topography, and rows of “constant” or “variable”). This would make it much easier to follow the comparisons. In addition, comparisons between model fits to data are fundamentally qualitative in how they are currently presented, making it difficult to know if the conclusion reached by the authors is quantitatively supported by their modeling. Another column could be added to the encouraged additional table for misfit to various datasets, like strain, stress orientation, and GPS measurements, even if they only represent relative misfits compared to each other rather than to true values, since many of these are only weakly constrained.

Response: Thank you so much for your kind words regarding our work! The comments really encouraged us. We have modified the table in the supplementary materials (supplementary table S1) to include details of our model setups and quantitative comparisons between our model prediction and observation. We hope this revised table with quantitative evaluations could make our conclusions easier to be assessed. In addition, we have included several sections in the revised “Methods” section to describe how we did the quantitative evaluations between the model prediction and observation.

Other issues in clarity exist due to semantic definitions (e.g., L84 “the mantle and lithospheric dynamics”). I assume mantle convection was meant here since mantle can also be lithosphere. Another example is broadly using basal traction to apparently refer to any mantle convective force between the asthenosphere and lithosphere, rather than purely referring to surface-parallel forces). In a framework of understanding contributions of difference forces, clearly defining these forces is important. This would also help with the fact that the authors really don’t make any clear conclusions about the relative contributions of body forces, basal tractions, or plate boundary

forces, rather simply stating in the discussion that the lithospheric thickness gradients are important. It isn't clear which force lithospheric thickness gradients should be associated with (it seems that this would affect both body forces and viscous drag, or the authors preferred term, "basal tractions"). Thus, the authors never clearly tie their introduction with their results and discussion.

Response: Thanks for the comments. We agree that the terminology in our original manuscript was loosely defined, and how the lithospheric thickness variations contribute to the driving forces was not clear. In the revised manuscript, we have added a paragraph describing our definitions of different forces when the realistic 3D lithospheric structure is considered (lines 803-835). In the same paragraph, we also discussed how lithospheric thickness variations impact the proposed driving forces. In addition, we have substantially revised our manuscript to strengthen the link between our "Introduction" and "Results and Discussion" sections. Particularly, we have added a quantitative evaluation of the relative importance of three proposed driving forces in driving the WUS crustal deformation (Fig. 9).

Regarding the methods, I think there are some unclear points. The authors state that they create a 3D viscosity model "adjusted to match the observed direction patterns of crustal stress". This seems very circular, as they then go on to say that these viscosity variations are important in controlling crustal stress orientations, which of course makes sense if they were derived from the stress orientations themselves. I think it is important for the authors to clearly identify what is an input to their model vs what is an output. This is exemplified by the L520 "One notable feature along the lithospheric step is the presence of a long weak zone within the crust". Here, the authors make it sound like a model result, when it is clearly embedded in the input model.

Response: Thank you for the constructive comments! Our preferred lithospheric effective viscosity structure was converted from a recent seismic attenuation study (Hearn, 2021, GJI). Seismic attenuation represents the anelastic properties of rocks (Romanowicz & Mitchell, 2015, Treatise on Geophysics; Takei, 2017, AREPS). Therefore, the seismic attenuation map determines the spatial pattern of effective viscosity (Fig. R1) but leaves the absolute value undetermined. In practice, we followed the same approach in Liu & Hasterock (2018, Science) to convert seismic attenuation to effective viscosity. The best-fitting conversion coefficients were determined by matching the predicted maximum horizontal compression (S_{Hmax}) directions to the observation

(Levandowski et al., 2018, Nature Geoscience; Lund Snee & Zoback, 2020, Nature Communications). In the revised manuscript, we have deleted all statements about the impact of the effective viscosity structure on crustal stress to avoid circular arguments.

[Figure Redacted]

Surface deformation patterns, such as the GPS-measured surface motion and deformation rate, are the goal of our prediction. Therefore, the observed surface deformation patterns were not used to construct the effective viscosity structure. Although most previous studies took GPS or geological measurements as known surface kinematic conditions to calculate depth-integrated lithospheric effective viscosity structure (e.g., Flesch et al., 2000, Science; Ghosh and Holt, 2012, Science; Gosh et al., 2013, JGR; Bahadori et al., 2021, Nature Communications), Ghosh et al. (2013, JGR) showed that this approach could not fully reproduce the input GPS measurements. This observation calls for an alternative approach that is independent of GPS measurements to construct the three-dimensional (3D) effective viscosity structure, such as our approach. We also tested the sensitivity of predicted surface motion to the effective viscosity structure, and it only shows weak sensitivity in the plausible regions. Therefore, the inclusion or exclusion of geodetic measurements in the construction process does not impact the best-fitting effective viscosity structure.

In practice, the long crustal weak zone was revealed and converted from the seismic attenuation map showing high attenuation in this region (Fig. R1). We have revised line 520 (lines 1224-1225 in the revised manuscript) to clarify that this crustal weak zone is in the input viscosity structure, which is converted from seismic attenuation.

The wording in the original “Methods” section and some places in the main text are confusing. Therefore, we have substantially revised the “Methods” section and the main text to clarify how we construct the lithospheric effective viscosity structure and deleted all statements about the impacts of the effective viscosity structure on crustal stress.

In conclusion, the takeaway of this paper is important, but the strength of their article is not clear in its current form due to the lack of quantitative comparisons between the different models (and therefore, plate driving forces, which they discuss quite a bit in the Introduction to set up their problem). Also, it is extremely difficult to keep track of all the models and their different assumptions, inputs and results. Lastly, and perhaps most importantly, the lack of clarity in writing and terminology will prevent this manuscript from having the impact that it likely could have given the importance of the result. My recommendation is that a significant amount of manuscript revision needs to be done to make the paper more intelligible, as it is extremely difficult to read and understand in its current form.

Responses: We really appreciate your constructive insights. We have substantially revised the manuscript for greater clarity and added more quantitative evaluations of our model prediction to better support our conclusions. Particularly, we have added quantitative comparisons between our model prediction and observation to show the effects of different parameters tested in this study, and the parameters and results are listed in the supplementary table S1. In addition, we have introduced a quantitative evaluation of the relative importance of three different proposed driving forces in driving crustal deformation (Fig. 9). In the revised supplementary table S1, we have also included detailed setups of all models. We hope the revision has a better logical flow and can easily be assessed.

Detailed Comments:

Abstract

L21: “Particularly, the interaction between the mantle flow and lithospheric thickness step along the eastern boundary of Basin and Range represents a key driving mechanism for localized intracontinental deformation.” Given that the motivation in the Intro is revealing the contributions of plate boundary forces, lithospheric body forces, and viscous drag to deformation, which category does this fall under? I could see this being a lithospheric body force (normal force exerted on LAB topography from flow) and/or a viscous drag. It’s not clear to me that this study actually illuminates the contributions between the stated body forces. I think the motivation should be changed to clearly apply to the conclusions, or the conclusions should be put in the context of the motivation.

Responses: Thank you for bringing this up! We have changed the abstract and introduction part to show that our motivation is exploring the impacts of complex 3D lithospheric structures on crustal deformation and stress state, as it is an unexplored but important aspect. The interaction between the mantle flow and lithospheric thickness step will primarily change the contribution of mantle convection to crustal deformation by changing the magnitude of crustal stress caused by the plate boundary effect and basal tractions.

L26: uplifting -> uplift

Responses: We have revised this sentence and the word “uplifting” has been deleted.

L26-28: The first part of this sentence is grammatically unclear. I assume you mean that geological and geophysical evidence show clear signs of extension.

Responses: Yes, here we mean the geological evidence shows clear signs of long-term extension. We have revised this sentence for greater clarity (lines 48-50).

L33-34: This sentence needs to be linked clearly to the stated 3 categories of driving forces. Alternatively, you could walk back the focus on the 3 categories, and simply state that you are exploring something that is largely unexplored in the context of these numerical models.

Responses: Thanks for this suggestion. We have revised this sentence to show that the complex 3D lithospheric structure affects all three proposed driving forces, and its effects were unexplored in previous studies (lines 55-57).

L42-44: There has been a lot of research done on the controlling forces of this change in plate motion and is largely attributed to plate boundary forces. An example paper exploring these forces should be cited (perhaps a geodesy paper).

Response: We have revised this paragraph to acknowledge previous geodetic studies and a citation was added in line 83.

Fig. 1: I recommend adding the boundaries of the Northern, Central, and Southern Basin and Range provinces since you discuss them later in the paper. You should also cite the reference for LAB depth.

Response: Thank you for the suggestions! We have added the boundaries of the northern, central, and southern Basin and Range (B&R) based on Jones et al. (1992, Tectonophysics). The LAB depth is a hybrid model constructed upon LITHO 1.0 and a body-wave tomography model (Schmandt & Lin, 2014, GRL). We have added details about how we constructed this LAB depth model in the revised "Methods" section (lines 1737-1744).

L68: Need to clarify that PNW volcanoes are not intraplate. The main edifices are typical calc-alkaline subduction-related volcanoes and shouldn't be included with intraplate volcanoes.

Response: This is a great point, and we really appreciate it. In the revised manuscript, we have deleted all volcanism-related content according to another reviewer's suggestion.

L83-84 and 112: Could you clarify what you mean by "one-way" coupling? Seems like a contradiction. Does "one-way" coupling mean all forces are drag from asth to lithosphere? What effect does "two-way" coupling actually have? It seems that all it implies is that it allows for more spatial complexity in your model.

Response: This is a good question! The one-way coupling approach only allows the convecting mantle to exert horizontal shear along the flat LAB. This horizontal shear is usually calculated in a global mantle convection model only considering long-wavelength deep density anomalies. Notably, the lithosphere, which has a laterally varying thickness in reality, does not influence the mantle flow in this approach. The scenario modeled by one-way coupling may not be realistic in regions with sharp lateral gradients in lithospheric thickness, such as the WUS. Besides, our recent high-resolution mantle convection model shows that lateral variations in lithospheric thickness prominently affect the shallow mantle flow beneath the WUS, and such interaction is critical to reproduce the observed flow-induced SKS splitting pattern (Zhou & Liu, 2018, EPSL). However, this lithospheric effect on mantle flow is neglected in the studies using the one-way coupling approach. We have revised this part to point out the meaning and defect of one-way coupling in lines 120-126.

In our fully dynamic model, the lithosphere and convecting mantle are allowed to interact naturally. For example, the lithospheric thickness step along the eastern boundary of B&R blocks the shallow mantle flow in our model (Fig. 3). Besides, the shallow mantle flow pushes the thick cratonic lithosphere through normal stress along the lithospheric thickness step (Fig. 7). This natural coupling allows more spatial complexity in our model, and it is more realistic.

L84, L97, L105: Recommend changing from "lithospheric and mantle dynamics" to "lithospheric and mantle convective dynamics" or "crustal and mantle dynamics"

Response: Thanks for the suggestion! We have changed the expression to "lithospheric and convective mantle dynamics" for better clarity. In addition, we have changed all similar expressions throughout the manuscript.

L130-L138: There don't seem to be any geological constraints described in this paragraph. All are geophysical.

Response: We have removed the word "geologically" (line 407).

L140: This language makes it very difficult to understand what is a constraint and what is a result. Do you force your model to fit topographic changes? How do you do this as a function of time? What do you mean your model reproduces volcanism?

Response: This is a good point. Our convective mantle structure is adopted from a time-dependent mantle flow model using the hybrid inversion scheme (Starting from 20 Ma, Zhou & Liu, 2017, G-cubed). The predicted time-dependent mantle flow field was used to investigate the spatial correlation between the intruding hot asthenospheric material and surface volcanism in the WUS (Zhou & Liu, 2018, Nature Geoscience), calculate mantle-flow induced asthenospheric anisotropy (Zhou et al., 2018, EPSL), and quantify the dynamic topography changes since Miocene (Zhou & Liu, 2019, EPSL). The results show great agreements between the model prediction and independent geological and geophysical observations. Since these contents are beyond the scope of this manuscript, we invite the reviewer and readers to check our previous publications for details. We have rewritten this part to clearly show what are constraints, what are results, and what are done in previous studies (lines 407-440).

L149-150: Isn't the slab from seismic tomography, rather than consistent with it? How does it get modeled as segmented?

Response: In our convecting mantle structure, there are two main slabs, namely the Juan de Fuca (JdF) slab and the Farallon slab. These two slabs were reproduced by different approaches (line 410-413). The young JdF slab was reproduced by the forward data-assimilation approach (Zhou & Liu, 2017, G-cubed; Liu & Stegman, 2011, EPSL). The agreement between the model-predicted and seismically imaged segmented JdF slab independently validates our approach. The segmentation of the JdF slab is caused by the dynamic pressure gradient across the subducting slab (Liu & Stegman, 2011, EPSL). The ancient Farallon slab, which now is located beneath the central

and eastern US, was reproduced through the adjoint approach (Zhou & Liu, 2017, G-cubed; Liu & Gurnis, 2008, JGR), which takes the seismic tomography image as a reference present-day state.

Fig 3: It would be very helpful to quantify the fit between SKS and predicted flow if you are going to use this as justification for a "better" model.

Response: The quantitative comparison between the predicted and observed SKS splitting patterns was done in our previous work (Zhou et al., 2018, EPSL). Here, we want to show that recently imaged depth-dependent azimuthal anisotropy (Zhu et al., 2020, JGR) is consistent with our predicted mantle flow as an independent constraint. In the revised manuscript, we have added a quantitative comparison between the observed azimuthal anisotropy direction and predicted active mantle flow fields (defined as velocity larger than 0.01 cm/yr in the North American Plate fixed reference frame). The predicted active mantle flow with a flat LAB exhibits a larger average angular misfit (49.7°) than the model with a spatially varying LAB (37.5°).

In addition, why should azimuthal anisotropy (an absolute measure of deformation/flow) correlate with predictions relative to the NA reference frame? I know this comparison is done a lot in geodynamic modeling studies, but it isn't clear to me why it would be valid, as the data and results are in different reference frames.

Response: This is a great question! Most mantle flow models are constructed in a lower mantle fixed reference frame, in which the lithosphere and convective mantle are both moving relative to the lower mantle. Since the surface plate motion could be modeled as a rigid rotation to the first order, it is easy to change the lower mantle fixed reference frame to a surface plate fixed reference frame. To achieve the reference frame transformation, we applied a rigid rotation, which does not involve any deformation, to both the lithosphere and convective mantle. The resulting mantle flow in a surface plate fixed reference frame measures the total relative motion between the surface plate and the underlying convective mantle, showing the absolute deformation. Therefore, comparing the observed azimuthal anisotropy and the predicted mantle flow in a surface plate fixed reference frame is valid.

L211-216: This sentence seems out of place and should probably be in the Introduction/Motivation section. It doesn't seem to particularly contribute to the discussion in this paragraph. Also might help with clarity if it is broken into multiple sentences.

Response: Thanks for the suggestion, we have moved this part to the “Introduction” section and rewritten it for greater clarity (lines 334-345).

L226: dominant-dominated

Response: We have changed this word accordingly (line 732).

Fig. 4: Very nice conceptual example of the effects of the different parameterizations. I would recommend this difference be quantified somehow to show the point quantitatively. Also, change caption (b) for clarity: “a lithosphere of constant thickness but varying Moho depth” rather than “same as in M1”

Response: Thank you for this comment! We have performed a seismicity prediction power analysis to quantitatively compare the predicted crustal deformation rate and the observed seismicity. Besides, we have included a vectorial analysis between the predicted and observed surface motion. The methodologies are detailed in the revised “Methods” section (lines 1908-1933). We also have changed the description of M2 accordingly to improve clarity.

L260: This section title (“Effects of the heterogeneous lithosphere on crustal deformation”) is trivially different from the prior section (“Mantle and crustal deformation controlled by the complex lithospheric structure”), neither of which really clearly state the parameters you are going to test in these sections.

Response: We have substantially revised the “Results and Discussion” section, particularly we have renamed these two subsections to “Complex Lithospheric Structure Controls WUS Crustal Deformation” and “WUS Crustal Deformation due to Different Driving Mechanisms”.

Again, clearly defining the differences in the models through a table or some other relatively simple figure would greatly help the clarity of which models are testing which contributions.

Response: Thanks for the suggestion! We have modified the supplementary table S1 to include the differences in the models and all quantitative evaluations of these models.

L285: May be worth noting briefly how this viscosity is constrained since it seems to have a significant effect (and/or referencing the methods)

Response: We have added a brief description of how we construct this effective viscosity structure (lines 448-450) and a reference to “Methods” in this sentence. We have also expanded the “Methods” section to provide details about the lithospheric effective viscosity structure (lines 1811-1856).

L311: Which lithospheric layers are varying? Both crustal thickness and total lithospheric thickness? Also how does this vary from M3? Clearly the results are different but it isn't easy to see how/why it varies from M3

Response: Both crustal and total lithospheric thickness have lateral variations in M6. Compared to M5, which has a lateral varying Moho depth, the difference is in the lithospheric mantle thickness. Compared to M3, the difference is the lithospheric density structure and the density anomalies in the convective mantle. In M6, the continental lithosphere is treated as a viscous layer without internal density variations, and the convective mantle does not have any density anomalies either. While in M3, we include all density anomalies within the continental lithosphere and convective mantle. In M6, we solely estimate the plate boundary effect in driving crustal deformation, accounting for the realistic 3D lithospheric viscosity structure. To better present the differences among all the models, we have included the details of model setups in the revised supplementary table S1.

L319: This seems contrary to your results (Fig.5b,c). The model that has variable viscosity but a uniform lithosphere (M5) reproduces very similar predicted strain magnitudes to the varying lithospheric structure (M6). Perhaps quantifying misfit would make your point more clear, but it

certainly isn't qualitatively clear that it makes a big difference in stress and strain predictions (although seems to make a difference for GPS predictions)

Response: This is a great point! Here, we want to emphasize the deformation and surface motion changes in the Colorado Plateau (CP) and the Wyoming Plateau (WP), where the lithosphere is significantly thicker than the rest of WUS. In M5, where the CP and WP have the same lithospheric thickness as the rest of WUS, crustal deformation diffuses into their interiors (Fig. 5b). While lateral variations in lithospheric thickness are considered (M6), the crustal deformation is localized along the lithospheric thickness step on the thin lithosphere side. Besides, M6 has a larger seismic prediction power than M5, indicating improvements in predicted crustal deformation. We have revised this part to clarify our point (lines 978-980).

Another important aspect in differentiating M5 and M6 is their respective predicted crustal motion, where the latter shows a notably better alignment with GPS observation. This further strengthens the role of the variable LAB depth.

L326: Should this be a new subsection? Frankly, it isn't clear in this paper when we are transitioning from a Section to a Subsection, and these should be clearly indicated in the section headers.

Response: Thank you for the comment! We have reorganized the "Results and Discussion" section and divided into five subsections with clear and specific focuses.

L338-340: Are you talking about only the black arrows here? Should point that out in the reference to the figures (e.g., cf. 5d, 6d, black arrows)

Response: Here, we are talking about the magenta arrows, which show the model predictions. We have added the reference to the figures in the revised manuscript (lines 903, 939, and 985).

L367: I also find the comparison interesting between M7 and M8. We can clearly see that the magnitude of surface motion is strongly linked to crustal rheology, however the orientations

remain nearly identical. However, this comes back to the fact that is glossed over in the methods, where your viscosity field is derived from surface motions. Regardless, this is another place that would benefit from a quantitative analysis rather than purely qualitative comparison.

Response: We have added a quantitative comparison between the predicted and observed surface motion. The results show that M8 has a significantly smaller average magnitude of surface velocity residue than M7, while the angular misfit remains similar (lines 969-971). Our effective viscosity structure was a constrained a priori converted from a recent seismic attenuation map. The observed surface motion and crustal deformation rate were not used to construct the lithospheric effective viscosity. Instead, they are the goal of our prediction. Please see the details in our response above.

L377-380: This statement is unclear to me. It sounds like the significance of the interaction is that it is not significant given no change in local stress. Maybe change to “variations in the viscosity of the convective mantle clearly plays a secondary role relative to LAB gradients in controlling crustal stresses”. Given this, it would imply that viscous drag (or at least basal traction) is a negligible plate driving force in the interior. This would be good to link to the overall discussion on driving forces.

Response: Thanks for your constructive comment! We have revised this sentence (lines 1137-1140) to reflect that lateral gradients in LAB depths are critical in localized crustal deformation along the lithospheric thickness step and leave the discussion of driving forces to the “Revised Role of Different Forces in Driving WUS Crustal Deformation” section. In the revised manuscript, we quantitatively evaluate the relative importance of the proposed driving forces (Fig. 9). The result shows that basal tractions play the least important role in deforming the interior of B&R.

L380-382: May be worth labeling this on the figures? I don't see particularly large changes in deformation rate between the two models.

Response: We have revised this sentence to specifically point out that basal tractions locally increase the crustal deformation rate in the northern RM and along the eastern boundary of northern B&R (lines 1140-1143).

L386-389: I assume these results are for M9? I think it would help to explicitly state somewhere which model these are the predictions for. I assume M9 but it isn't stated. It is hard to keep track of all these models and their differences.

Response: These results are from M3, our best-fitting model. We labeled this in the caption of Fig. 7. This paragraph seems out of place in the original manuscript. In the revised manuscript, we have separated this part into a subsection discussing the mechanism of localized intraplate deformation (lines 1187-1262). Besides, we have also clarified which model is used to present the locally enhanced basal tractions along the lithospheric thickness step.

Figure 7: State which model this is for: M7? This is also a place where an explicit definition of traction would be helpful. I would assume that (a-c) is the traction parallel to the horizontal plane, but there are clear upward and downward tractions being observed, which I would think should be called normal forces. Again, maybe a semantical difference.

Response: This is from M3, our best-fitting model, as labeled in the figure title (the bold part). Traction is defined as the force vector on a plane with a unit area. Therefore, the radial component shown in (a) – (c) as colored background is the normal force in the radial direction, and the vectors show the shearing forces parallel to the plane.

L435: Change “lithospheric thickness and strength” to “lithospheric properties”. As you modify crustal thickness as well as lithospheric thickness, the former statement doesn't clearly show all the factors you tested. The “properties” is more vague.

Response: This is a good suggestion! We have changed this part accordingly (line 1265).

L440: “pressure drag” is this a basal traction? or a body force? This isn't made clear.

Response: This is part of basal tractions, which was largely ignored in previous studies. Specifically, this “pressure drag” is due to the significant positive dynamic pressure beneath the thin lithosphere region caused by the thick lithospheric keel blocking the eastward asthenospheric

flow. We have specified this in the revised manuscript (lines 1197-1199). In addition, we have incorporated a paragraph showing our definitions of the forces in lines 803-835.

L440: Should this be Fig 7? It isn't stated that lithosphere changes at all in Fig 8.

Response: Yes, this should be Fig. 7. M10 (shown in the original Fig. 8, now Fig. 6) has a seismically inferred 3D lithospheric thickness and effective viscosity structures, as well as a geophysically constrained 3D density structure. We have included all the parameters of all models in the revised supplementary table S1.

L447: Should this be Fig 9a?

Response: No, the gradients of lithospheric GPE is originally shown in Fig. 8a (Fig. 6a in the revised manuscript).

L460-461: This probably should be cited if you are going to state this. Given the long active tectonic history of the western half of North America, many of these faults are reactivated from prior tectonic events and can rupture even if they are not presently at the optimal orientation relative to the driving forces since they will be weaker than unbroken rock.

Response: This is a good point. Recent geophysical estimations of the WUS crustal stress state (Levandowski et al., 2018, Nature Geoscience; Lund Snee and Zoback, 2020, Nature Communications) revealed patterns similar to that depicted in focal mechanism solutions. They exhibit similar directional patterns and faulting regimes across the WUS. While some local variations exist in the focal mechanism solutions, the overall pattern appears to be reasonably robust. We believe that while oriented weak planes might be important on a local scale, their influence seems to be a second-order effect on a regional scale. We have added comparisons to the geophysically estimated crustal stress pattern in the revised manuscript (Fig. 8).

L465-468: Many authors attribute the N-S compression to SAF-related tectonic forces as well as JdF plate boundary forces. This should probably be added. How can you decouple plate boundary forces from basal tractions in strike-slip regions?

Response: This is a good question! In this study, we focus on the regional pattern of crustal deformation instead of individual faults. Hypothetically, pure strike-slip faults, if all with vertical orientations, seem more sensitive to horizontal forcing like plate boundary forces than buoyancy forces that act more in the vertical direction. However, in practice, basal tractions, although originated from buoyancy force of the convecting mantle, can be oriented in the horizontal direction as well. Then the separation of the two (plate boundary force vs tractions) could take a similar approach as that presented in this study: Taking advantage of the flexibility of data assimilation, we could isolate the effects of different dynamic components. For example, we first perform a simulation with the independently constructed 3D lithospheric effective viscosity structure and prescribed plate motions (M6). In this model, all density anomalies in the continental lithosphere and convective mantle are excluded. Therefore, the lithospheric body forces and basal tractions due to density-driven mantle flow are not included. The resulting crustal deformation and stress state solely reflect the plate boundary effect. Then, we add in all density anomalies in the convective mantle and perform a second simulation that includes plate boundary effect and basal tractions (M9). By calculating the difference between these two models, we isolate the crustal stress caused by basal tractions. Similarly, we can isolate the crustal stress due to lithospheric body forces. By comparing the magnitude of crustal stress caused by plate boundary effect, basal tractions, and lithospheric body forces, we can evaluate their relative importance (Fig. 9), even in the central and southern B&R, where plate boundary effects and basal tractions result in the same faulting regime (Fig. 5c and 5f).

L470-: A couple things here:

L468-473: How does combining N-S extensional stress with W-E extensional stress lead to W-E extension? This seems to overstate the importance of basal traction given that it doesn't seem to alter GPE predictions. 2) Where are you talking about when you say southern Basin and Range? The southern Basin and Range has a relatively specific location (central to southern Arizona and Mexico; see Jones et al 1992), but there are only 2 or 3 focal mechanisms in this area, only 1 of

which seems to be N-S oriented. 3) I would bet that including a model of lithospheric properties in Mexico (such as the subducting slab and LAB variations) would significantly alter the results in the southernmost portion of this model (away from the SAF). This should probably be acknowledged and likely makes the discussion on what factor is controlling stress orientation at these southern latitudes only loosely constrained by measurements in the US.

Response: We appreciate your insightful comments. First, the crustal stress resulting from plate boundary effect exhibits strong W-E extension and weak N-S compression (Fig. 5c), while the crustal stress due to basal tractions generally exhibits N-S extension (Fig. 5f). The combination of these two results in N-S extension in the model considering all three driving forces (Fig. 9a). The E-W extension caused by lithospheric body forces (Fig. 6a) intensifies the E-W extension in the southern B&R. We have also clarified this in lines 1429-1435.

We have added boundaries separating the B&R into the northern, central, and southern segments in the revised figures (based on Jones et al., 1992, Tectonophysics). The southern B&R we are talking about is the same as in Jones et al. (1992, Tectonophysics). In our model, we include part of the deforming northern Mexico (north of 20 °N). We assume that northern Mexico has the same lithospheric properties (i.e., layered structure of density and effective viscosity) as the B&R. The Moho and LAB depths are derived from CRUST 1.0 and LITHO 1.0. Besides, the comparison between a regional model and a global model shows that the mantle flow fields beneath this region are similar (Peng and Liu, 2023, ESR). Therefore, we argue that our model captures the first-order dynamics in this region.

We agree that the number of focal mechanism solutions is limited in the southern B&R, so we further added recent geophysical estimations of maximum horizontal compression directions (S_{Hmax}) and $A\phi$ parameters, which reflects the faulting regime, from Lund Snee & Zoback (2020, Nature Communications). This dataset provides additional constraints in the southern B&R. Our predicted crustal stress field agrees with both datasets (Fig. 8). Overall, we think our model reproduces the dynamics in the southern B&R relatively well.

L494-496: But which of the 3 forces is this? You set up a discussion that states that this study contributes to answering which dominates the WUS, but never explicitly link the results to the forces.

Response: Here, the lithosphere-asthenosphere interaction impacts basal tractions and plate boundary effect (Fig. 5). We have deleted this paragraph in the revised manuscript since it is confusing and redundant to the new section “Mechanism for Localized Intraplate Deformation”. In the revised manuscript, we have added definitions of different driving forces (lines 803-835) and a quantitative evaluation of their relative importance in the WUS (Fig. 9).

L508-509: definition of GPE -> do you mean the magnitude of GPE? The definition of GPE doesn't change, it is the values included in the calculation that change.

Response: Here, we want to refer to the magnitude of GPE and the unphysical assumption involved in its traditional calculation (e.g., Ghosh et al., 2012, Science; Bahadori et al., 2019, Nature Communications). We have added a paragraph discussing the traditional way of lithospheric GPE calculation and the simplifications involved in it (lines 824 – 829) to clarify our point.

L509-512: remove also: it seems that the big change between how the lithosphere and asthenosphere act IS through changing the lateral viscosity structure, and thus ease of asthenospheric convection/motion.

Response: We respectfully disagree with this suggestion. Our analysis demonstrates that lateral variations in the effective viscosity structure is critical in reproducing the observed crustal deformation. Moreover, we also show that the lithospheric effective viscosity features, particularly the narrow crustal weak zone, are ultimately controlled by the E-W contrasting lithospheric structure. In the revised manuscript, we have formed a section discussing the formation mechanism for localized lithospheric deformation and its relation to the rheological features and local lithospheric thinning (lines 1222-1262).

L516: I recommend using a phrase besides “transition zone” for clarity, like the "transition from thin to thick lithosphere". The transition zone has definitions to the geophysical community

(mantle transition zone) as well as the tectonic community in the western US (transition zone is a small tectonic strip between southern B&R extension and the Colorado Plateau). Both of these are pretty aseismic and may just confuse readers.

Response: Thanks for the suggestion! We have revised the manuscript, and this sentence has been removed from the manuscript.

L520: This feature is embedded in your model through your parameterization and the depth to the Moho and LAB. However, you state it here like it is a result of your modeling rather than an input. You should either delete this sentence (the 2nd sentence is the important one for discussion anyway) or modify it to clearly differentiate results from inputs.

Response: We have modified this sentence to reflect that the crustal weak zone is a feature embedded in our input effective viscosity structure (lines 1224-1225).

L521: change "may reflect" to "its rheology is controlled by thermal, compositional, and mechanic properties." Clearly it must be controlled by these, as the weak zone in your crust is embedded through your own parameterization of the model.

Response: Thanks for the suggestion! We have changed the wording accordingly (lines 1224-1225). Regarding the crustal weak zone in the input effective viscosity structure, it is based on a recent seismic attenuation map (Hearn, 2021, GJI) revealing high attenuation along the eastern boundary of B&R (Fig. R1). The high attenuation zone should physically reflect the low effective viscosity in this region. Please see the details in our response above. We have expanded the "Methods" section to detail how we construct the effective viscosity structure (lines 1811-1856).

L549-554: I do not think that the widespread presence of the low velocity zone necessitates non-vertical transport of melt. In your previous paragraph you conclude that (and I agree) that translithospheric extension is likely necessary to get asthenospheric melt to the surface. Given the rather small volume of most (non-Yellowstone related) magmatism in the WUS, why couldn't these localized extension regions only tap the asthenosphere below them, while melt away from these zones remains trapped below an impermeable lithosphere due to no vertical strain?

Response: We agree that most localized extension regions typically tap the asthenosphere right below them. We also agree that melt away from trans-lithospheric extension zones may be trapped below an impermeable LAB. Here, by non-vertical transport, we want to highlight the intraplate volcanism along the southern boundary of CP. These volcanoes are situated inside the CP but not directly above the widespread low-velocity zone. Instead, they correlate well with areas of rapid crustal deformation. This alignment implies that non-vertical transport may happen beneath the southern edge of CP.

In addition, I don't think it is appropriate to compare mostly small-volume basaltic (i.e., very low viscosity) magmatism that spatially dominates the intraplate WUS to a long-lived rhyolitic (i.e., high viscosity) magmatism, and perhaps even less appropriate to compare locations of short-lived asthenosphericly-sourced magmatism to long-lived magma residence in magma reservoirs. I agree that below more silicic magmatic centers, there is no need for the reservoir to lie below the edifice, as the edifice location is likely controlled by brittle, active structures in the upper crust (especially for high viscosity magmas). There is plenty of literature on that. But I do not think this applies to pathways for very low viscosity melts. Also, you are implying that melt is able to be taps from 100s of km away from where they erupt with this comparison, whereas active magmatic systems generally show reservoirs below edifices, while offset, tend to be no more than ~10 km away from the edifice (Lerner et al., 2020). This mix increase with depth, but it would be quite a stretch to say that the center of the B&R is sourcing magmatism near the Wasatch. Lastly, a simple calculation comparing the volume of the LVZ below the B&R with some assumed melt percentage with the total erupted volume in the B&R will show that very little assumed melt reaches the surface.

You state a more plausible interpretation in L554-556, by saying melt reaches the surface if lithospheric stresses allow for it. In a more compressional state, the lithosphere could probably be inferred to act as an impermeable cap, even if melt is present.

This could all be corrected by simply deleting the sentence encompassed by L553-554

Response: We really appreciate your comments! We agree that the mechanism for long-lived rhyolitic magmatism may significantly differ from that for small-volume basaltic magmatism. As stated above, we think most of the surface magmatism in the WUS should tap through trans-lithospheric extension zones and sample the asthenosphere right below them. Besides, the non-vertical transport may only occur in some places, such as around the southern edge of CP. In the revised manuscript, we have deleted all contents related to intraplate volcanism according to another reviewer's suggestion.

L560-561: It would be nice to have more specific examples rather than just pointing out large regions. I think the East African Rift system near cratons and the Anatolian system are both areas where literature has interpreted that magmatism correlates with throughgoing lithospheric structures and presumed changes in LAB depth.

Response: We really appreciate this comment. In the revised manuscript, we have deleted all volcanism-related material according to another reviewer's suggestion.

Methods:

L607: This is a really low-resolution model of crustal thickness. Why not use better constrained thicknesses, like from Schmandt et al 2015 (model on Earthscope EMC page)

Response: We compared the Moho depth in CRUST 1.0 and Schmandt et al. (2015, GRL). Generally, they agree with each other in the WUS (Supplementary Fig. 5). However, our model includes parts of Canada and Mexico, where Schmandt et al. (2015, GRL) does not provide Moho depth data. Consequently, we selected the Crustal 1.0 as our starting model.

L627-629: This seems circular...Of course adding 3D viscosity variations to your model in this study leads to better fits to crustal stresses, the viscosity model is derived from fitting the stresses!

Response: This study focuses on the crustal deformation in the WUS. Since 3D viscosity variations significantly impact surface deformation (e.g., Humphreys & Coblenz, 2007, ROG; Liu &

Hasterock, 2018, Science), an independently constructed 3D effective viscosity structure is required. In our approach, the 3D effective viscosity structure is constrained a priori by matching the directional pattern of crustal stress. Observed surface deformation patterns, such as GPS-measured surface motion and deformation rate, were not used to construct the effective viscosity structure. Instead, they were the goal of our prediction. We have revised the “Methods” section to clarify that the effective viscosity structure was converted from the seismic attenuation map, and the best-fitting conversion coefficients were determined by matching the directional pattern of crustal stress (lines 1811-1856). Please see the details in our response above. In addition, we have removed all statements about the impacts of effective viscosity on crustal stress from the main text.

REVIEWER COMMENTS

Reviewer #1 (Remarks to the Author):

Summary

This manuscript is modified from one that I reviewed previously.

Recommendation

The authors have addressed most of the issues that I raised, which led to improvements to the manuscript. I would consent to the publication of the paper provided that they adequately address the issues detailed below.

Detailed Review:

Despite this version's improved organization, I still find it difficult to understand the relationship between the many models that the authors present by only reading the main text. I thus suggest moving Table S1 to the main text because it can greatly improve the clarity of the paper. Besides, in their response, they claim to have changed "Empty" in Table S1 to "None" following my suggestions, but this is not true. "None" or "Not included" are more appropriate than "Empty".

How the maps in Fig. 9 are computed are unclear to me. Lines 499–501 gives a brief description, but the sentence is very difficult to understand. It says that they computed the ratios of the magnitudes of the three types of driving forces, and the magnitudes are equivalent to the second invariant of the stress tensor. My first question is: why the magnitudes are equivalent to the stress invariants? Second, how did the authors separate the contributions of the three driving forces to the stress field? Third, against what reference values are the ratios computed?

I don't see the point of presenting the model M10. This issue contributes to my general impression that the paper lacks a clear focus: It seems to me merely a laundry list of all findings, big and small, from an exhaustive parameter study, whereas I believe that papers suitable for high-impact journals such as Nature Communications should tell a cohesive story with a clear focus. Both the other reviewer and I have alluded to this in our reviews, but the authors seem to be only willing to make limited changes. I personally find this frustrating and strongly suggest that the authors further trim the paper by removing redundant materials.

Lines 454–475: I also find this paragraph redundant because the focus of the paper is, at least from my reading it, how the complex lithospheric structures in the WUS control the intraplate deformation in the region, NOT the origins of these complex structures. I thus suggest deleting this paragraph.

Although the authors claim that they have fixed all grammatical errors and typos, I still found a

few, which are listed below. There are likely more of them, so I urge the authors to carefully proofread their manuscript again.

Line 213: "implying for" is grammatically incorrect. "for" should be deleted.

Line 360: Should "Fig. 5e" be "Fig. 5h" instead?

Line 363: It's better to replace "considering" with "adding...to".

Line 388: "incorporates" is more appropriate than "considers".

Line 373: What "mantle convection" means here is unclear. The difference between M6 and M9 and the other models seem to be the heterogeneous lithospheric structures, so "heterogeneous lithospheric structures" may be more appropriate here.

Lines 514–515: The "and" between "solutions" and "geophysically" should be replaced with a comma.

Fig. 5: Why does M9 seem to have worse GPS misfits than M6? Why do Southern California and the area near the Mendocino Triple Junction have significantly higher GPS misfits consistently in all scenarios?

The authors use the expression "X MPa" in the legends of a few figures, which looks very much like a typo at first glance. I suggest adding a legend for each subplot instead.

M3 should be labeled in Figs. 7 and 8 so that readers can easily see what models they are. Why is "with" italic in the caption of Fig. 7?

Fig. 8b: The bars are difficult to see. Consider increasing their thickness.

Reviewer #2 (Remarks to the Author):

Overall, I think the authors have done well in addressing my major concerns on the clarity of the manuscript. I very much appreciate their statistical quantification of the results, and still strongly suggest they find a way to implement this table in some form into their manuscript (perhaps just adding the quantitative measures of fit to some portion of the model result figures would be sufficient). I also suggest they add a Conclusion section where they concisely wrap up their major conclusions from their thorough search through the importance of the different tested parameters. I have some minor comments below, but find this manuscript suitable for publication once these are addressed.

Below are responses to the researchers' corrections, with original comments in black, their response in blue, and my follow-up comments in red. Comments that are deleted from prior review

have been sufficiently addressed

L33-34: This sentence needs to be linked clearly to the stated 3 categories of driving forces.

Alternatively, you could walk back the focus on the 3 categories, and simply state that you are exploring something that is largely unexplored in the context of these numerical models.

Responses: Thanks for this suggestion. We have revised this sentence to show that the complex 3D lithospheric structure affects all three proposed driving forces, and its effects were unexplored in previous studies (lines 55-57).

L34: Should change "affects" to "is linked to"

L42-44: There has been a lot of research done on the controlling forces of this change in plate motion and is largely attributed to plate boundary forces. An example paper exploring these forces should be cited (perhaps a geodesy paper).

Response: We have revised this paragraph to acknowledge previous geodetic studies and a citation was added in line 83.

The way this statement is phrased makes it feel out of place. Perhaps link this statement to which of the 3 forces are affected by the plate boundary forces mentioned, and say that while true, this region also appears linked to broader deformation throughout the WUS

Fig. 4: Very nice conceptual example of the effects of the different parameterizations. I would recommend this difference be quantified somehow to show the point quantitatively. Also, change caption (b) for clarity: "a lithosphere of constant thickness but varying Moho depth" rather than "same as in M1"

Response: Thank you for this comment! We have performed a seismicity prediction power analysis to quantitatively compare the predicted crustal deformation rate and the observed seismicity. Besides, we have included a vectorial analysis between the predicted and observed surface motion. The methodologies are detailed in the revised "Methods" section (lines 1908-1933). We also have changed the description of M2 accordingly to improve clarity.

This section is now much more clear. The addition of the quantification of fit shows clearly how important the different parameters are in helping fit the observed data. I might recommend adding those numbers to some part of the Fig. 4 images (top right of the maps?)

L260: This section title ("Effects of the heterogeneous lithosphere on crustal deformation") is

trivially different from the prior section (“Mantle and crustal deformation controlled by the complex lithospheric structure”), neither of which really clearly state the parameters you are going to test in these sections.

Response: We have substantially revised the “Results and Discussion” section, particularly we have renamed these two subsections to “Complex Lithospheric Structure Controls WUS Crustal Deformation” and “WUS Crustal Deformation due to Different Driving Mechanisms”.

Again, I don't think these titles are clearly pointing out the differences in the Discussion points. They sound pretty much the same to me, with one just being more detailed. Would it be better to title these something like "The role of lithospheric thickness on deformation in the WUS" for the 1st section and "The role of different driving forces on reproducing WUS deformation"? They don't need to be these titles, but I recommend you think long and hard about the major takeaway of each section, and choose your title after you figure that out.

Again, clearly defining the differences in the models through a table or some other relatively simple figure would greatly help the clarity of which models are testing which contributions.

Response: Thanks for the suggestion! We have modified the supplementary table S1 to include the differences in the models and all quantitative evaluations of these models.

I still recommend that you try to fit this into the main text.

L282-285 of new manuscript

"assuming complete compensation ... at 100 km depth". You make a good point in why this may not be appropriate, but it would also be worth pointing out that the lithosphere is thicker than 100 km in some parts of your study area (e.g., the Colorado Plateau). Given that isostasy relies on a rigid body over a fluid medium, the compensation depth is implied to be the LAB on Earth. Clearly, a compensation depth of 100 km depth is not valid when the rigid body (lithosphere) is thicker than this. I recommend pointing that out as well in the text

L274- 302 in revised manuscript:

It seems the current last 2 paragraphs of "Complex lithospheric structure" section should be first paragraphs for the "WUS driving mechanisms" section, since they really provide the motivation for performing these other 7 tests. I recommend moving them to the driving mechanisms section because I originally thought that these were results from the first 3 tests rather than motivations

for the other tests.

L319: This seems contrary to your results (Fig.5b,c). The model that has variable viscosity but a uniform lithosphere (M5) reproduces very similar predicted strain magnitudes to the varying lithospheric structure (M6). Perhaps quantifying misfit would make your point more clear, but it certainly isn't qualitatively clear that it makes a big difference in stress and strain predictions (although seems to make a difference for GPS predictions)

Response: This is a great point! Here, we want to emphasize the deformation and surface motion changes in the Colorado Plateau (CP) and the Wyoming Plateau (WP), where the lithosphere is significantly thicker than the rest of WUS. In M5, where the CP and WP have the same lithospheric thickness as the rest of WUS, crustal deformation diffuses into their interiors (Fig. 5b). While lateral variations in lithospheric thickness are considered (M6), the crustal deformation is localized along the lithospheric thickness step on the thin lithosphere side. Besides, M6 has a larger seismic prediction power than M5, indicating improvements in predicted crustal deformation. We have revised this part to clarify our point (lines 978-980).

Another important aspect in differentiating M5 and M6 is their respective predicted crustal motion, where the latter shows a notably better alignment with GPS observation. This further strengthens the role of the variable LAB depth.

Again, I recommend adding the quantitative metrics of fit onto the maps themselves so the reader can quickly see that the fits improve (rather than needing to look very carefully for differences in the results themselves). This could be done on Figure 5. I also recommend having the reproduced velocity and deformation rate maps next to each other for the models (e.g., 1row = M4-M6 defm rate, 2row = M4-M6 velocity residual, 3rd row M7-9 defm rate, 4th row M7-9 velocity residual).

Also change "residue" to residual in color bar

Fig 5 caption: change empty to "none" or "no convecting mantle" per R1's comments

L440: "pressure drag" is this a basal traction? or a body force? This isn't made clear.

Response: This is part of basal tractions, which was largely ignored in previous studies.

Specifically, this "pressure drag" is due to the significant positive dynamic pressure beneath the thin lithosphere region caused by the thick lithospheric keel blocking the eastward asthenospheric flow. We have specified this in the revised manuscript (lines 1197-1199). In addition, we have

incorporated a paragraph showing our definitions of the forces in lines 803-835.

I think you should explicitly state that pressure drag is part of the basal traction force, as you define it. Especially given that pressure usually implies a body force.

This paper needs a Conclusion section! You that to really push your "take home" point for the reader. After reading about the results of so many different models, it would be nice to tie them all together into the major takeaways for the reader. The less work the reader has to do to remember the important points, the better the paper will be received!

Response letter

We greatly appreciate the time and effort the reviewers devoted to providing valuable and insightful feedback on our work. The constructive comments and suggestions are beneficial in enhancing the quality of our manuscript. In response to the feedback, we have tried our best to incorporate all the suggestions and comments, with tracked changes in the revised manuscript. Once again, we sincerely appreciate the reviewers' constructive inputs to our work. We are committed to addressing any further concerns the reviewers may have.

Below are our detailed point-to-point responses to the reviewers' comments and suggestions, with original comments in black and our response in blue.

To reviewer #1:

The authors have addressed most of the issues that I raised, which led to improvements to the manuscript. I would consent to the publication of the paper provided that they adequately address the issues detailed below.

Response: Thank you for the insightful suggestions. In this revision, we have reorganized the sections presenting our numerical results and reduced the number of models presented in the main text. Besides, we have modified the manuscript to emphasize the effect of the lithospheric thickness step on forming localized intraplate crustal deformation, and moved all discussions related to the effect of crustal rheology to the Methods section. We hope the modifications will make our manuscript more cohesive and clearer.

Detailed Review:

1. Despite this version's improved organization, I still find it difficult to understand the relationship between the many models that the authors present by only reading the main text. I thus suggest moving Table S1 to the main text because it can greatly improve the clarity of the paper. Besides, in their response, they claim to have changed "Empty" in Table S1 to "None"

following my suggestions, but this is not true. “None” or “Not included” are more appropriate than “Empty”.

Response: Thanks for the suggestions! We have moved Supplementary Table 1 to the main text. We apologize for the inappropriate wording in Supplementary Table 1. We have changed the word “empty” to “none” in Table 1 and all other related places in the text and captions.

To make our manuscript more concise and easier to understand, we have moved four models about the effects of 3D crustal effective viscosity structure to the Methods section. In the revised main text, we use three models (M1-M3) to illustrate the importance of the complex lithospheric structure in reproducing the observed crustal deformation, and three more models (M4-M6) to estimate the magnitude of different driving forces and associated crustal deformation by accounting for the geophysically inferred 3D heterogeneous lithosphere.

2. How the maps in Fig. 9 are computed are unclear to me. Lines 499–501 gives a brief description, but the sentence is very difficult to understand. It says that they computed the ratios of the magnitudes of the three types of driving forces, and the magnitudes are equivalent to the second invariant of the stress tensor. My first question is: why the magnitudes are equivalent to the stress invariants? Second, how did the authors separate the contributions of the three driving forces to the stress field? Third, against what reference values are the ratios computed?

Response: These are good questions! Since stress is a second-order tensor, it is hard to measure its magnitude directly. A typical way is using the second invariant of the stress tensor as a scalar measure of its magnitude (e.g., Bahadori et al., 2022, Nature Communications), similar to using the second invariant of the strain rate tensor to show the deformation rate.

We take advantage of the flexibility of data assimilation to separate the contributions of different driving forces. For example, we first perform a simulation with the independently constructed 3D lithospheric effective viscosity structure and prescribed plate motions (M4). In this model, all density anomalies within the continental lithosphere and convective mantle are excluded. Therefore, the lithospheric body forces and basal tractions due to density-driven mantle flow are not included in M4. The resulting crustal deformation and stress state solely

reflect the plate boundary effect. Then, we add in all density anomalies in the convective mantle and perform a second simulation that includes the plate boundary effect and basal tractions (M5). By calculating the difference between these two models, we estimate the crustal stress caused by basal tractions. Similarly, we can estimate the crustal stress due to lithospheric body forces. In the revised manuscript, we have explained how we calculate the crustal stress caused by basal tractions (lines 365-382) and lithospheric body forces (lines 613-614).

To evaluate the relative importance, we calculate the ratios of the stress magnitude due to different driving forces. In this calculation, we do not use any reference value. The ratio is calculated using the stress magnitudes shown in Fig. 8a-8c. For example, to calculate the ratio shown in Fig. 8d (T_{LBF}/T_{BT}), we divided T_{LBF} shown in Fig. 8a by T_{BT} shown in Fig. 8b. This method has been widely used in previous studies to show the relative importance of different driving forces (e.g., Bahadori et al., 2022, Nature Communications).

3. I don't see the point of presenting the model M10. This issue contributes to my general impression that the paper lacks a clear focus: It seems to me merely a laundry list of all findings, big and small, from an exhaustive parameter study, whereas I believe that papers suitable for high-impact journals such as Nature Communications should tell a cohesive story with a clear focus. Both the other reviewer and I have alluded to this in our reviews, but the authors seem to be only willing to make limited changes. I personally find this frustrating and strongly suggest that the authors further trim the paper by removing redundant materials.

Response: Thanks for the comment. We have substantially reorganized the sections presenting our numerical results in this revision. Particularly, we have moved the models discussing the role of crustal rheology to the Methods section and focused our discussion on the effect of the lithospheric thickness step in generating localized intraplate deformation. Now, in the "Results and Discussion" section, we first show the importance of the 3D lithospheric structure in reproducing the observed surface deformation pattern, then we estimate the driving forces and associated crustal deformation with a focus on the effect of the lithospheric thickness step, and finally we discuss the relative importance of different driving forces.

In this revision, we have kept M10 (M6 in the revised manuscript) since this model shows the effect of lithospheric body forces arising from the lateral gradients of lithospheric gravitational potential energy (GPE). The lithospheric body force was proposed as an important driving force for crustal deformation (e.g., Jones et al., 1998, *Nature*; Flesch et al., 2000, *Science*; Ghosh et al., 2012, *Science*). However, previous studies always estimated this force with simplified lithospheric structures assuming lithospheric isostasy and a global compensation depth (e.g., Flesch et al., 2000, *Science*; Ghosh et al., 2012, *Science*; Becker et al., 2015, *Nature*; Bahadori et al., 2022, *Nature Communications*). Here, we want to emphasize that a lithospheric mantle, which is denser than the ambient asthenosphere and has a spatially varying thickness, is important to reproduce the surface topography and lithospheric GPE. Therefore, we chose to keep this model to demonstrate the relative importance of different driving forces in deforming the WUS crust.

4. Lines 454–475: I also find this paragraph redundant because the focus of the paper is, at least from my reading it, how the complex lithospheric structures in the WUS control the intraplate deformation in the region, NOT the origins of these complex structures. I thus suggest deleting this paragraph.

Response: Thanks for this suggestion! In the revision, we have moved this paragraph to the Methods section. According to our analysis, the crustal weak zone is important in reproducing the fast-deforming Intermountain Seismic Belt (ISB). Also, it may result from the long-term interaction between the active asthenospheric flow and the thick lithospheric root. Therefore, we decided to keep this paragraph to complete the discussion on the forming mechanism of the ISB. We agree that this paragraph may not be directly related to the focus of the main text, so we have moved it to the Methods section.

Line 213: “implying for” is grammatically incorrect. “for” should be deleted.

Response: We have deleted the word “for” (line 241).

Line 360: Should “Fig. 5e” be “Fig. 5h” instead?

Response: Thanks for pointing this out. Here the reference should be “Fig. 5h”. In this revision, we substantially revised this section. Now this sentence is in the Methods section and the reference is updated to “Supplementary Fig. 5f” in line 1144.

Line 363: It’s better to replace “considering” with “adding...to”.

Response: We have changed the word to “adding ... to” and moved this paragraph to the Methods section (line 1147).

Line 388: “incorporates” is more appropriate than “considers”.

Response: We have changed the word accordingly (line 496).

Line 373: What “mantle convection” means here is unclear. The difference between M6 and M9 and the other models seem to be the heterogeneous lithospheric structures, so “heterogeneous lithospheric structures” may be more appropriate here.

Response: In the revision, this sentence has been deleted from the main text.

Lines 514–515: The “and” between “solutions” and “geophysically” should be replaced with a comma.

Response: We have added a comma in this sentence (lines 819-820).

Fig. 5: Why does M9 seem to have worse GPS misfits than M6? Why do Southern California and the area near the Mendocino Triple Junction have significantly higher GPS misfits consistently in all scenarios?

Response: Yes, M9 (M5 in the revised manuscript) has a worse GPS fit than M6 (M4 in the revised manuscript) due to the eastward motion driven by the eastward asthenospheric flow beneath the WUS. We want to emphasize that these two models represent two end-member cases, one for the plate boundary effect only and the other for the plate boundary effect plus basal tractions. Neither of them match the observation well since both lack the effect of lithospheric body forces.

Our models do not predict surface motion well near the Mendocino Triple Junction. The triple junction may have complex rheology that affects the coupling among the Pacific plate, Juan de

Fuca plate, North American plate, and the underlying convective mantle. In our models, the lithospheric effective viscosity structure near the triple junction may not capture all the relevant rheological features. Therefore, we leave this tectonically complex region for future studies that may require a more complex rheological law.

The authors use the expression “X MPa” in the legends of a few figures, which looks very much like a typo at first glance. I suggest adding a legend for each subplot instead.

Response: Thanks for this suggestion! We have added a separate legend for each map showing the stress and surface velocity in Figs. 4, 5, and 6.

M3 should be labeled in Figs. 7 and 8 so that readers can easily see what models they are. Why is “with” italic in the caption of Fig. 7?

Response: We have added label M3 in Figs. 7 and 8 (Figs. 6 and 7 in the revised manuscript). The italic font is a typo, and we have corrected it in the revised manuscript.

Fig. 8b: The bars are difficult to see. Consider increasing their thickness.

Response: Thanks for the suggestion. We have increased the bar thickness in Fig. 8b (Fig. 7b in the revised manuscript).

To reviewer #2:

Overall, I think the authors have done well in addressing my major concerns on the clarity of the manuscript. I very much appreciate their statistical quantification of the results, and still strongly suggest they find a way to implement this table in some form into their manuscript (perhaps just adding the quantitative measures of fit to some portion of the model result figures would be sufficient). I also suggest they add a Conclusion section where they concisely wrap up their major conclusions from their thorough search through the importance of the different tested parameters. I have some minor comments below, but find this manuscript suitable for publication once these are addressed.

Response: Thanks for your kind words! In this revision, we have moved Supplementary Table 1 to the main text and added key statistics to the maps showing comparisons between model prediction and observation. Since Nature Communications does not permit a separate Conclusion section, we alternatively added a paragraph and a figure (Fig. 10 in the revised manuscript) concluding our key findings at the end of the manuscript.

Detailed Review:

L34: Should change "affects" to "is linked to"

Response: We have changed the word here accordingly (lines 33-34).

L44-47: The way this statement is phrased makes it feel out of place. Perhaps link this statement to which of the 3 forces are affected by the plate boundary forces mentioned, and say that while true, this region also appears linked to broader deformation throughout the WUS.

Response: In this revision, we have rephrased this statement to point out that the interpretation from geodetic studies is consistent with surface kinematics, but the underlying dynamics remains unclear (lines 43-54).

This section is now much more clear. The addition of the quantification of fit shows clearly how important the different parameters are in helping fit the observed data. I might recommend adding those numbers to some part of the Fig. 4 images (top right of the maps?)

Response: Thanks for the suggestion! We have moved Supplementary Table 1 to the main text and added key statistics for each map in revised Figs. 4 and 5 to better present the quantification of fits to observation.

Again, I don't think these titles are clearly pointing out the differences in the Discussion points. They sound pretty much the same to me, with one just being more detailed. Would it be better to title these something like "The role of lithospheric thickness on deformation in the WUS" for the 1st section and "The role of different driving forces on reproducing WUS deformation"? They don't need to be these titles, but I recommend you think long and hard about the major takeaway of each section, and choose your title after you figure that out.

Response: Thank you for the suggestion! In the revision, we have reorganized the section "WUS Crustal Deformation due to Different Mechanisms". In the revised manuscript, we have moved the effects of crustal rheology to the Methods section and focused our discussion on estimating driving forces and associated deformation with a focus on the effect of the lithospheric thickness step in this section. Thus, we have changed the title to "Estimations of Different Driving Forces and Associated Deformation".

I still recommend that (Supplementary Table 1) you try to fit this into the main text.

Response: Thanks for this recommendation. We have moved Supplementary Table 1 to the main text.

L282-285 of new manuscript

"assuming complete compensation ... at 100 km depth". You make a good point in why this may not be appropriate, but it would also be worth pointing out that the lithosphere is thicker than 100 km in some parts of your study area (e.g., the Colorado Plateau). Given that isostasy relies on a

rigid body over a fluid medium, the compensation depth is implied to be the LAB on Earth. Clearly, a compensation depth of 100 km depth is not valid when the rigid body (lithosphere) is thicker than this. I recommend pointing that out as well in the text.

Response: This is an excellent point! We have added this point in lines 323-324 and lines 327-329.

L274- 302 in revised manuscript:

It seems the current last 2 paragraphs of "Complex lithospheric structure" section should be first paragraphs for the "WUS driving mechanisms" section, since they really provide the motivation for performing these other 7 tests. I recommend moving them to the driving mechanisms section because I originally thought that these were results from the first 3 tests rather than motivations for the other tests.

Response: Thanks for this recommendation! We have moved the last two paragraphs of the "Complex lithospheric structure" section to the beginning of the driving mechanism section (lines 309-337).

Again, I recommend adding the quantitative metrics of fit onto the maps themselves so the reader can quickly see that the fits improve (rather than needing to look very carefully for differences in the results themselves). This could be done on Figure 5. I also recommend having the reproduced velocity and deformation rate maps next to each other for the models (e.g., 1row = M4-M6 defm rate, 2row = M4-M6 velocity residual, 3rd row M7-9 defm rate, 4th row M7-9 velocity residual). Also change "residue" to residual in color bar

Response: Thanks for this suggestion. In this revision, we have reorganized this subsection and Fig.5. In the revised Fig. 5, the key statistics has been added to the maps. We have also applied similar modifications to all maps showing simulation results. Additionally, we have changed the word "residue" to "residual" throughout the manuscript.

Fig 5 caption: change empty to "none" or "no convecting mantle" per R1's comments

Response: We have changed the word “empty” to “none” in Fig. 5’s caption, Table 1 (originally Supplementary Table 1), and all related places in the text in the revised manuscript.

L440: I think you should explicitly state that pressure drag is part of the basal traction force, as you define it. Especially given that pressure usually implies a body force.

Response: This is a good point. We have added some words to specify this point in line 696.

This paper needs a Conclusion section! You that to really push your "take home" point for the reader. After reading about the results of so many different models, it would be nice to tie them all together into the major takeaways for the reader. The less work the reader has to do to remember the important points, the better the paper will be received!

Response: Thank you for this suggestion. Since Nature Communications does not permit a separate Conclusion section, we have alternatively added a concluding paragraph and a conclusion figure to summarize our key findings at the end of the manuscript (lines 885-920).

REVIEWERS' COMMENTS

Reviewer #1 (Remarks to the Author):

Recommendation

I was not expecting to review this manuscript again because in the last round of review, I said I would consent to the publication of the manuscript provided that the authors address my comments. Nonetheless, the editor insisted on asking me to review it for a third time, suggesting that she also has concerns over the work. I confess that I was very reluctant to do so because I had already spent so much time on it, and that the experience of reading the manuscript is far from being pleasant, mostly due to its poor organization, about which I have complaint many times. Nonetheless, out of the sense of responsibility to the community, I decided to take up the unpleasant task once again. However, I was soon shocked by how disrespectful the authors are to my time: Table 1, which was brought to the main text upon my request and is instrumental in clarifying their bewildering parameter-test procedure, contains two IDENTICAL rows (5 and 10)! I read the two rows multiple times to make sure that I didn't hallucinate because I simply could not believe that such an obvious typo could appear in such an important place in a manuscript that has already gone through two rounds of review. This incident, together with the many typos and grammatical errors that I caught in the previous two rounds, suggests to me that the authors are extremely negligent, which inevitably leads one to doubt the reliability of their work. I am also furious about their complete lack of respect for my precious time spent in painstakingly reviewing their work. I thus refuse to review the manuscript again unless I receive the guarantee that my time spent in doing so will be respected.

Response letter

We greatly appreciate the reviewer's time and effort in providing feedback and suggestions on our work. In response to the feedback, we have corrected the remaining problems in the manuscript and reviewed all the text and display items carefully and thoroughly. Once again, we sincerely appreciate the reviewer's constructive input to our work.

Below are our detailed point-to-point responses to the reviewers' comments and suggestions, with original comments in black and our response in blue.

To reviewer #1:

I was not expecting to review this manuscript again because, in the last round of review, I said I would consent to the publication of the manuscript, provided that the authors address my comments. Nonetheless, the editor insisted on asking me to review it for a third time, suggesting that she also has concerns over the work. I confess that I was very reluctant to do so because I had already spent so much time on it, and that the experience of reading the manuscript is far from being pleasant, mostly due to its poor organization, about which I have complaint many times. Nonetheless, out of the sense of responsibility to the community, I decided to take up the unpleasant task once again. However, I was soon shocked by how disrespectful the authors are to my time: Table 1, which was brought to the main text upon my request and is instrumental in clarifying their bewildering parameter-test procedure, contains two IDENTICAL rows (5 and 10)! I read the two rows multiple times to make sure that I didn't hallucinate because I simply could not believe that such an obvious typo could appear in such an important place in a manuscript that has already gone through two rounds of review. This incident, together with the many typos and grammatical errors that I caught in the previous two rounds, suggests to me that the authors are extremely negligent, which inevitably leads one to doubt the reliability of their work. I am also furious about their complete lack of respect for my precious time spent in painstakingly reviewing their work. I thus refuse to review the manuscript again unless I receive the guarantee that my time spent in doing so will be respected.

Response:

We sincerely apologize for the remaining problem in the manuscript and thank you for pointing this out. In the last revision, we reduced the number of models presented in the main text and changed the order of models, which led to the remaining problem. In this revision, we have carefully examined all text and display items and corrected the problem in Table 1.

We truly appreciate your precious time and effort in providing constructive comments and suggestions to improve our manuscript's quality. Again, we sincerely apologize for all the typos and grammatical issues in previous versions of our manuscript, and we express our most sincere gratitude for your time and effort devoted to helping us improve the manuscript.